# A Little Depth Goes a Long Way:
# The Expressive Power of Log-Depth Transformers

**William Merrill**[*]
Allen Institute for AI
willm@allenai.org

**Ashish Sabharwal**
Allen Institute for AI
ashishs@allenai.org

## Abstract

Recent theoretical results show transformers cannot express sequential reasoning problems over long inputs, intuitively because their computational *depth* is bounded. However, prior work treats the depth as a constant, leaving it unclear to what degree bounded depth may suffice for solving problems over short inputs, or how increasing the transformer's depth affects its expressive power. We address these questions by analyzing transformers whose depth can grow minimally with context length $n$. We show even highly uniform transformers with depth $\Theta(\log n)$ can express two important problems: *recognizing regular languages*, which captures state tracking abilities and was known to be expressible only by an unconventional, non-uniform model of transformers, and *graph connectivity*, which underlies multi-step reasoning. Notably, both of these problems cannot be expressed by fixed-depth transformers under standard complexity conjectures, demonstrating the expressivity benefit of growing depth. Moreover, our theory quantitatively predicts how depth must grow with input length to express these problems, showing that depth scaling is more efficient than scaling width or chain-of-thought steps. Empirically, our detailed experiments designed to bridge the expressivity vs. learnability gap reveal that our theoretical depth requirements for regular language recognition closely match the practical depth requirements for successfully training transformers. Thus, our results clarify how depth affects a transformer's reasoning capabilities, and provide practical guidance for effective depth selection for sequential reasoning.

## 1 Introduction

A line of recent work analyzing the intrinsic computational power of transformers, the neural architecture behind today's immensely successful large language models (LLMs), has established that, with fixed depth, transformers cannot express many simple problems outside the complexity class $\mathsf{TC}^0$, including recognizing regular languages and resolving connectivity of nodes in a graph (Merrill and Sabharwal, 2023a; Chiang et al., 2023). These problems conceivably underlie many natural forms of sequential reasoning, such as state tracking (Liu et al., 2023; Merrill et al., 2024) and resolving logical inferences across long chains (Wei et al., 2022). Thus, these results suggest inherent limitations on the types of reasoning transformer classifiers can perform. Yet, these findings come with an important caveat: even if transformers cannot solve such problems exactly for inputs of arbitrary lengths, they may still be able to solve them over inputs *up to some bounded length*. This perspective, coupled with the fact that *treating depth as fixed* is crucial to prior analyses placing transformers in $\mathsf{TC}^0$, motivates two related questions about depth as an important resource for a transformer, in relation to the context length over which it reasons:

1. **Bounded Context:** If fixed-depth transformers cannot theoretically express certain problems over unbounded context lengths, can they still express them over bounded but still practically

---

[*]Work partially conducted as a PhD student at New York University.

39th Conference on Neural Information Processing Systems (NeurIPS 2025).

"large enough" contexts? Can we quantitatively characterize the context length up to which transformers are effective for different problems as a function of their depth?

2. **Dynamic Depth:** Can minimally scaling the depth of a transformer allow it to solve such problems for arbitrarily long inputs? How does this compare in efficiency to scaling width (i.e., model dimension) or scaling inference-time compute via chain-of-thought steps?

We address these questions by analyzing the expressive power of "universal" transformers (also called "looped" transformers) where a fixed model is given dynamic depth by repeating a block of middle layers a variable number of times (Dehghani et al., 2019; Yang et al., 2024; Geiping et al., 2025). We capture the regime where depth grows minimally with context length, with the middle layers repeated $\Theta(\log n)$ times on contexts of length $n$. We prove that even such highly uniform transformers, when allowed log-depth, can recognize regular languages[2] and solve graph connectivity, two important reasoning problems known to be beyond fixed-depth transformers (Merrill and Sabharwal, 2023a). Our core technical contribution enabling this is Lemma 1, that fully uniform transformers can compute *division* and *remainder* of small integers. This not only obviates the need for non-uniformity and special positional encodings relied upon in prior work, it is also an interesting finding on its own.

Our result has two interesting interpretations. First, it directly shows that, by **dynamically increasing their depth** as $\Theta(\log n)$ on inputs of length $n$, one can construct transformers to solve regular language recognition (Theorem 1) and graph connectivity (Theorem 2) for arbitrary context length. In contrast, chain-of-thought (CoT) steps, used for additional test-time compute by newest LLMs such as OpenAI o1 (OpenAI, 2024) and DeepSeek-R1 (DeepSeek AI, 2025), must be scaled superlogarithmically (Theorem 4) to solve these problems, and width must be scaled superpolynomially (Theorem 3), as shown in Figure 1. Thus **scaling depth** more efficiently allows solving these reasoning problems compared to scaling width or using CoT.

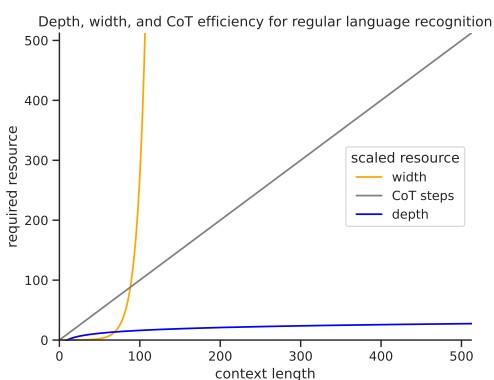

Depth, width, and CoT efficiency for regular language recognition

Figure 1: To recognize a regular language over inputs of length $n$, the depth of a universal transformer can grow $\Theta(\log n)$ by Theorem 1. On the other hand, width must grow *superpolynomially* (Theorem 3), and the number of chain-of-thought steps must be *superlogarithmic* (Theorem 4). The precise depth and width coefficients plotted here were **obtained experimentally** in Section 7.

Second, a universal transformer unrolled to a fixed (independent of input size) depth $d$ is a special case of a standard $d$-depth transformer, namely one with a highly uniform structure (parameters shared across layers). Thus, our result shows that standard transformers with a *fixed depth $d$* can recognize regular languages (Corollary 1.2) and solve graph connectivity problems (Corollary 2.1) as long as one cares only about *bounded inputs* of size $2^{O(d)}$. This allows us to quantify how many layers are necessary for a desired input size. For instance, Corollaries 2.1 and 1.2 imply that with a depth of only 32, such as in LLaMA 3.1 7B (Meta AI, 2024) and OLMo 7B (Ai2, 2024), transformers can recognize regular languages up to strings of length 107 and solve connectivity for graphs with up to 128 vertices. A depth of 80 (such as in LLaMA 3.1 70B) makes these input size limits practically unbounded: strings of length up to 440K and graphs with up to 2.1B vertices. **Empirically**, our experiments supplementing our theoretical findings demonstrate that scaling depth as $\Theta(\log n)$ is necessary and sufficient for learning to recognize hard regular languages.[3]

We hope these findings serve as actionable guidance for practitioners to choose effective model depths for reasoning over long contexts, and motivate further exploration of dynamic depth as an inference-time compute strategy for transformer based LLMs.

---

[2]While Liu et al. (2023) provided a similar result, their construction relied heavily on an unconventional, *non-uniform* transformer architecture, requiring a different set of model weights for each input length. In contrast, our result holds in a stronger setting of highly uniform transformers—where model weights are not only fixed (independent of input length) as in practice but even shared across blocks of layers, enabling effective inference-time scaling. Our formal model incorporates standard architectural choices like residual connection and layer norm. We supplement our stronger theoretical findings with matching empirical learnability results.

[3]Code: `https://github.com/jopetty/word-problem/tree/willm/log-depth/log-depth`.

## 2 Preliminaries: Universal Transformers

We consider $(s, r, t)$-**universal transformers**, which are defined to have $s$ fixed initial layers at the start, a sequence of $r$ layers that is repeated some number of times based on the input length, and a sequence of $t$ fixed final/terminal layers. Thus, an $(s, r, t)$-universal transformer unrolled $d(n)$ times for input length $n$ has a total of $s + rd(n) + t$ layers. Geiping et al. (2025) empirically explored such transformers for scaling test-time computation for reasoning problems. A standard $d$-layer transformer is $(d, 0, 0)$-universal (equivalently, $(0, 0, d)$-universal), while a standard universal transformer (Dehghani et al., 2019; Yang et al., 2024) is $(0, 1, 0)$-universal.

**Definition 1.** A decoder-only $(s, r, t)$-universal transformer with $h$ heads, $d$ layers, model dimension $m$ (divisible by $h$), and feedforward width $w$ is specified by:

1. An embedding projection matrix $\mathbf{E} : \Sigma \to \mathbb{Q}^m$ and positional encoding function $\pi : \mathbb{N} \to \mathbb{Q}^m$, which we assume separates 1 from other indices (Merrill and Sabharwal, 2024);
2. A list of $s$ "initial" transformer layers (defined under "Transformer Sublayers" below);
3. A list of $r$ "repeated" transformer layers;
4. A list of $t$ "final" transformer layers;
5. An unembedding projection matrix $\mathbf{U}$ that maps vectors in $\mathbb{Q}^m$ to logits in $\mathbb{Q}^{|\Sigma|}$.

We next define how the transformer maps a sequence $w_1 \cdots w_n \in \Sigma^n$ to an output value $y \in \Sigma$; to do so, we will always specify that the transformer is **unrolled** to a specific depth function $d(n)$, which we will consider to be $d(n) = \lceil \log n \rceil$.[4] The computation is inductively defined by the **residual stream** $\mathbf{h}_i$: a cumulative sum of all layer outputs at each token $i$. In the base case, the residual stream $\mathbf{h}_i$ is initialized to $\mathbf{h}_i^0 = \mathbf{E}(w_i) + \pi(i)$. We then iteratively compute $s + rd(n) + t$ more layers, deciding which layer to use at each step as follows:

$$L^\ell = \begin{cases} s\text{-layer } \ell & \text{if } 1 < \ell \leq s \\ r\text{-layer } ((\ell - s - 1) \bmod r) + 1 & \text{if } s < \ell \leq s + rd(n) \\ t\text{-layer } \ell - s - rd(n) & \text{otherwise.} \end{cases}$$

We then compute $\mathbf{h}_1^\ell, \ldots, \mathbf{h}_n^\ell = L^\ell(\mathbf{h}_1^{\ell-1}, \ldots, \mathbf{h}_n^{\ell-1})$. The transformer output is a token determined by first computing the logits $\mathbf{h}_n^{\ell^*} \mathbf{U}$, where $\ell^* = s + rd(n) + t$, and then selecting the token with maximum score. We can identify a special token in $\Sigma$ with "accept" and say that a transformer **recognizes** language $L$ if, for every $w \in \Sigma^*$, it outputs "accept" if and only if $w \in L$.

An $(s, r, t)$-transformer unrolled to some fixed depth can be viewed as a "uniform" special case of a fixed-depth transformer. Thus, constructions of dynamic-depth transformers (depth $d(n)$ for inputs of length $n$) imply that, given any bounded context length $N$, there also exists a fixed-depth transformer with depth $d(N)$ for the task at hand. The fact that this can be done with a looped transformer with dynamic depth is, in fact, a stronger condition that shows the construction is uniform, which is formally important as non-uniform models of computation can have very strong and unrealistic power (cf. Merrill et al., 2022). In this way, our results about looped transformers will provide insights about standard, non-looped transformers with bounded context lengths.

**Transformer Sublayers.** To make Definition 1 well-defined, we will next describe the structure of the self-attention and feedforward sublayers that make up the structure of each transformer layer. Our definition of the transformer will have two minor differences from practice:

1. **Averaging-hard attention** (a.k.a., saturated attention): attention weight is split uniformly across the tokens with maximum attention scores.
2. **Masked pre-norm**: We assume standard pre-norm (Xiong et al., 2020) but add a learned mask vector $\mathbf{m} \in \mathbb{R}^m$ that can select specific dimensions of the residual stream per sublayer.

Under masked pre-norm, the sublayer input will be read as sequence of normalized residual stream values $\mathbf{z}_i = \mathsf{layer\_norm}(\mathbf{m} \odot \mathbf{h}_i)$, where $\odot$ is elementwise product and layer-norm can be standard layer-norm (Ba et al., 2016) or RMS norm (Zhang and Sennrich, 2019). The sublayer then maps $\mathbf{z}_1, \ldots, \mathbf{z}_n$ to a sequence $\delta_1, \ldots, \delta_n$, and updates the residual stream as $\mathbf{h}_i' = \mathbf{h}_i + \delta_i$.

**Definition 2** (Self-attention sublayer). The self-attention sublayer is parameterized by a mask $\mathbf{m} \in \mathbb{Q}^m$, output projection matrix $\mathbf{W} : \mathbb{Q}^m \to \mathbb{Q}^m$, and, for $1 \leq k \leq h$, query, key, and value matrices $\mathbf{Q}^k, \mathbf{K}^k, \mathbf{V}^k$, each of which is a projection from $\mathbb{Q}^m$ to $\mathbb{Q}^{m/h}$.

---

[4] Following computer science conventions, we let $\log n \triangleq \log_2 n$.

Given input $\mathbf{z}_i$, the self-attention sublayer computes queries $\mathbf{q}_i = \mathbf{Q}^k \mathbf{z}_i$, keys $\mathbf{k}_i = \mathbf{K}^k \mathbf{z}_i$, and values $\mathbf{v}_i = \mathbf{V}^k \mathbf{z}_i$. Next, these values are used to compute the attention head outputs:

$$\mathbf{a}_{i,k} = \lim_{\tau \to 0} \sum_{j=1}^{c} \frac{\exp(1/\tau \cdot \mathbf{q}_{i,k}^\top \mathbf{k}_{j,k})}{Z_{i,k}} \cdot \mathbf{v}_{j,k}, \quad \text{where } Z_{i,k} = \sum_{j=1}^{c} \exp\left(1/\tau \cdot \mathbf{q}_{i,k}^\top \mathbf{k}_{j,k}\right)$$

and $c = i$ for causal attention and $c = n$ for unmasked attention. The $\tau \to 0$ limit implements averaging-hard attention: all probability mass is concentrated on the indices $j$ for which the attention score is maximized. This idealization is similar to assuming the temperature of the attention is large relative to the sequence length $n$. Finally, the attention heads are aggregated to create an output to the residual stream $\delta_i = \mathbf{W} \cdot \text{concat}(\mathbf{a}_{i,1}, \dots, \mathbf{a}_{i,h})$.

**Definition 3** (Feedforward sublayer). The feedforward sublayer at layer $\ell$ is parameterized by a mask $\mathbf{m} \in \mathbb{Q}^m$ and projections $\mathbf{W} : \mathbb{Q}^m \to \mathbb{Q}^w$ and $\mathbf{U} : \mathbb{Q}^w \to \mathbb{Q}^m$.

A feedforward layer computes a local update to the residual stream via $\delta_i = \mathbf{U} \cdot \text{ReLU}(\mathbf{W}\mathbf{z}_i)$.

**Positional Encodings.** We will assume no positional encodings (Kazemnejad et al., 2023), but that there is a beginning of sequence (BoS) symbol at the start. As described by Merrill and Sabharwal (2024), our constructions will generalize to any positional encoding as long as either BoS is provided or the first token's positional encoding is linearly separable from all other positional encodings.

## 2.1 Memory Management in Universal Transformers

A technical challenge when working with universal transformers that add values to the residual stream is that if one is not careful, outputs from the previous iteration of a layer may interfere with its computation at a later iteration. This necessitates "memory management" of individual cells in which the transformer stores values. In particular, any intermediate values stored by a layer must be "reset" to 0 and any desired output values must be correctly updated after use in subsequent layers.

Appendix A discusses in detail how $\{-1, 0, 1\}$ values can be stored directly in the residual stream, while a general scalar $z$ can be stored either as $\psi(z) = \langle z, 1, -z, -1 \rangle$ in its *unnormalized form* or as the unit vector $\phi(z) = 1/\sqrt{2} \cdot \psi(z)/\sqrt{z^2+1}$ in its *normalized form* ("layer-norm hash"; cf. Merrill and Sabharwal, 2024). Importantly, however $z$ is stored, when it is read using masked pre-norm, we obtain $\phi(z)$. In Appendix A, we show how numerical values represented using $\psi$ or $\phi$ can be easily written (Lemma 4), read (Lemma 2), and deleted (Lemmas 5 and 6) from the residual stream. We will leverage these operations heavily in our theoretical constructions.

## 2.2 Numerical Datatype and Precision

Our constructions will involve working with scalars used as *pointers* to token positions, which will be stored in and retrieved from the residual stream (as discussed in Section 2.1). We thus need $\Omega(\log n)$ bits of precision. We now formalize the underlying datatype we use for our constructions.

We assume scalars are encoded as strings in $\{0,1\}^p$, with $p = c \log n$ for some fixed $c > 0$. Note that model parameters in our fully uniform setting cannot depend on $n$, but activations can. We assume there is some *datatype* $\mathbb{D}_p$ that assigns a numerical semantics for each string in $\{0,1\}^p$. For $x \in \mathbb{R}$, let $[x]_{\mathbb{D}_p}$ be $x$ rounded into $\mathbb{D}_p$, i.e., the bitstring whose numerical value in $\mathbb{D}_p$ is closest to $x$ (breaking ties in favor of the higher value).

Our constructions will be agnostic to the underlying details of $\mathbb{D}_p$. Instead, we will minimally assume that, for some fixed $c$, all arithmetic operations in the transformer computation graph (addition, multiplication, division, $\exp$, and layer-norm) are $p$-precise for $p \geq c \log n$ in the following sense:

**Definition 4** ($p$-Precise Operations). Let $f : \mathbb{R}^k \to \mathbb{R}$ be an operation with $p$-precision realization $\tilde{f} : \mathbb{D}_p^k \to \mathbb{D}_p$. We say $\tilde{f}$ is $p$-precise if, for any $x_1, \dots, x_k \in \mathbb{R}$ exactly representable in $\mathbb{D}_p$,

$$[f(x_1, \dots, x_k)]_{\mathbb{D}_p} = \tilde{f}([x_1]_{\mathbb{D}_p}, \dots, [x_k]_{\mathbb{D}_p}).$$

To apply this definition, we view the summation in attention heads as an $n$-ary operation. We also view layer-norm as a single operation from $\mathbb{R}^m \to \mathbb{R}^m$.

Definition 4 is naturally satisfied by the log-precision transformer model formalized by Merrill et al. (2024, Section 2.2 and Appendix A) and used in earlier work (Merrill and Sabharwal, 2023a,b), as long as sufficient precision is used internally to compute attention and layer-norm precisely. It is an open question how much precision is required by the internal primitive operations of summation and layer-norm to guarantee their output is $(c \log n)$-precise. We intentionally abstract away these low-level details here, especially because, in practice, additional precision is typically allocated in any case for attention and layer-norm computation (Micikevicius et al., 2018; Ai2, 2024).

Finally, we briefly note how Definition 4 relates to other datatype models. Chiang et al. (2023) propose a polynomial-precision rational datatype, which satisfies Definition 4 because the first $c \log n$ bits (and potentially more) are correct. In contrast, finite-precision transformers do not satisfy Definition 4 because only $O(1)$ bits are correct. In particular, while the mixed-precision model of Yang et al. (2025) can precisely represent attention, it cannot store $(c \log n)$-bit values in the residual stream or as the output of layer-norm. It therefore does not satisfy Definition 4 or suffice for our constructions.

# 3 Fixed Depth Transformers Can Divide Small Integers

A useful primitive for coordinating information routing in a log-depth transformer will be dividing integers and computing remainders. We therefore start by proving that transformers can perform integer division for small numbers, which will be a useful tool for our main results. Specifically, we show that given a non-negative integer $a_i$ no larger than the current position $i$, one can compute and store the (normalized) quotient and remainder when $a_i$ is divided by an integer $m$. This effectively means transformers can perform arithmetic modulo $m$ for small integers.

**Lemma 1** (Division). *Let $a_i, b_i, c_i, m \in \mathbb{Z}^{\geq 0}$ be such that $a_i = b_i m + c_i$ where $a_i \leq i$ and $c_i < m$. Suppose $\psi(i)$, $\psi(m)$, and $\phi(a_i)$ (or $\psi(a_i)$) are present in the residual stream of a transformer at each token $i$. Then, there exists a block of 7 transformer layers with causally masked attention and masked pre-norm that, on any input sequence, adds $\phi(b_i)$ and $\phi(c_i)$ to the residual stream at each token $i$.*

*Proof.* The overall idea is as follows. In the first layer, each position $i$ outputs an indicator of whether it's a multiple of $m$. It also adds $\phi(j)$ to the residual stream such that $j$ is the quotient $i/m$ if $i$ is a multiple of $m$. In the second layer, each position $i$ attends to the nearest position $j \leq i$ that is a multiple of $m$ and retrieves the (normalized) quotient stored there, which is $j/m = \lfloor i/m \rfloor$. It adds this (normalized) quotient in its own residual stream. We then use Lemma 7 (§A.3) to construct a third layer that adds $\phi(i - 1)$ and $\phi(i - 2)$ to the residual stream. A fourth layer checks in parallel whether the quotient stored at $i$ matches the quotients stored at $i - 1$ and $i - 2$, respectively. In the fifth layer, position $i$ counts the number of positions storing the same quotient as $i$, excluding the first such position. Finally, in the sixth layer, position $i$ attends to position $a_i$ to compute and add to the residual stream $\phi(\lfloor a_i/m \rfloor)$ (which is $\phi(b_i)$) and $\phi(a_i - m\lfloor a_i/m \rfloor)$ (which is $\phi(c_i)$). We next describe a detailed implementation of the construction, followed by an argument of its correctness.

Construction. The first layer uses the following attention head. The query at position $i$ is $q_i = \phi(i, m) = \phi(i/m)$ computed via Lemma 3 (§A.1) leveraging the assumption that $\psi(i)$ and $\psi(m)$ are present in the residual stream. The key and value at position $j$ are $k_j = v_j = \phi(j)$ Let $h_i^1 = \phi(j)$ denote the head's output. The feedforward sublayer computes $e_i = \mathbb{I}(h_i^1 = \phi(i/m))$ using Lemma 8 (scalar equality check, §A.4) on the first coordinate of $h_i^1$ and $\phi(i/m)$. By Lemma 9 (§B), $e_i = 1$ if and only if $i$ is a multiple of $m$ and, if $e_i = 1$, then $h_i^1 = \phi(i/m)$, i.e., it represents the quotient $i/m$. We store $h_i^1 = \phi(i/m)$ and $e_i$ to the residual stream.[5]

The second layer uses a head that attends with query $q_i = \langle 1, 1 \rangle$, key $k_j = \langle e_j, [\phi(j)]_0 \rangle$, and value $v_j = h_j^1$; both $e_j$ and $h_j^1$ can be read from the residual stream using masked pre-norm. This head attends to all positions $j \leq i$ that are multiples of $m$ (where $e_j = 1$), with $[\phi(j)]_0$, the first component of $\phi(j)$, serving as a tie-breaking term for breaking ties in favor of the *nearest* multiple of $m$. Lemma 10 (§B) shows this head outputs $h_i^2 = \phi(\lfloor i/m \rfloor)$, which we store in the residual stream.

The third layer uses Lemma 7 (§A.3) to add $\phi(i - 1)$ and $\phi(i - 2)$ to the residual stream at position $i$.

---

[5]As described in §8, a component will be added to the second layer to reset intermediate memory cells used in the first layer to 0 (this will happen analogously in later layers, but we will omit mentioning it).

In parallel for $k \in \{1, 2\}$, the fourth layer attends with query $q_i = \phi(i - k)$, key $k_j = \phi(j)$, and value $v_j = \phi(\lfloor j/m \rfloor)$ to retrieve the quotient stored at position $i - k$. It uses Lemma 8 (SA.4) on the first coordinate to store in the residual stream a boolean $b_i^k = \mathbb{I}(\phi(\lfloor i/m \rfloor) = \phi(\lfloor (i - k)/m \rfloor))$, indicating whether the quotient stored at $i$ matches the quotient stored at $i - k$.

In the fifth layer, position $i$ attends with query $q_i = \langle \phi(\lfloor i/m \rfloor), 1 \rangle$, key $k_j = \langle \phi(\lfloor j/m \rfloor), b_j^1 \rangle$, and value $v_j = 1 - b_j^2$. When the output $h_i^5$ of this layer is read through layer norm, it produces $\phi(h_i^5) = \phi(i \bmod m)$ as proved in Lemma 11 (§B).

The sixth layer attends with query $q_i = \phi(a_i)$, key $k_j = \phi(j)$, and value $v_j = \langle \lfloor j/m \rfloor, \phi(j \bmod m) \rangle$ (from layers two and five) to compute $\langle \phi(\lfloor a_i/m \rfloor), \phi(a_i \bmod m) \rangle$, which equals $\langle \phi(b_i), \phi(c_i) \rangle$.

The seventh and final layer cleans up any remaining intermediate values stored in the residual stream, setting them back to 0 as per Lemma 8. This is possible because all values $v$ are of the form $\phi(x)$ or a boolean, which means adding $-\phi(v)$ to the residual stream will reset the corresponding cell to 0. $\quad\square$

Our division construction is somewhat similar to the modular counting construction from Strobl et al. (2024), though the tools and underlying assumptions are different. Specifically, their approach relies on nonstandard position embeddings whereas ours uses masked pre-norm.

# 4 Log Depth Enables Recognizing Regular Languages

Constant-depth transformers cannot recognize regular languages, a natural task closely related to state tracking (Liu et al., 2023; Merrill et al., 2024). Liu et al. (2023, Theorem 1)[6] show that a variant of log-depth transformers can recognize regular languages using an associative prefix-sum construction (cf. Hillis and Steele Jr, 1986; Blelloch, 1990). However, it is, prima facie, unclear whether the *fully uniform* model of transformers we study—where the parameters cannot change at all with $n$—can implement their construction for two key reasons:

1. To handle information routing, Liu et al. (2023, Page 44) require parameters that are not fully uniform, meaning they can depend on the input length and depth. This leaves it unclear whether a *single transformer* (with a fixed set of parameters) could solve the task across *all* input lengths. In other words, their work leaves it unclear whether a single transformer could implement the approach in a way that generalizes to inputs of any length.
2. Liu et al. (2023) also make several simplifications to the transformer architecture: they add non-standard positional embeddings and remove residual connections and layer-norm. While one could adapt their construction to handle residual connections, it is not clear how to do this while also making their construction uniform, which requires proper memory management of cells in the residual stream (Section 2.1).

Our result, using Lemma 1, addresses both of these weaknesses. It shows, for the first time, that a single transformer with *fixed parameters* (w.r.t. input length $n$) can recognize strings of *any length*; moreover this transformer does not require specific positional encodings, allows for layer-norm (in fact leverages it), and allows for residual connections while remaining fully uniform.

**Theorem 1** (Regular Language Recognition). *Let $L$ be a regular language over $\Sigma$ recognized by a (non-)deterministic finite automaton with states $Q$. Let $\$ \notin \Sigma$. Then there exists a causally masked $(0, 8, 9)$-universal transformer with*

- *model dimension $m_{\mathrm{NFA}} = \mathrm{O}(|Q|^2)$, or $m_{\mathrm{DFA}} = \mathrm{O}(|Q| \log |Q|)$ if deterministic;*
- *feedforward width $w_{\mathrm{NFA}} = \mathrm{O}\big(2^{|Q|^4}\big)$, or $w_{\mathrm{DFA}} = \mathrm{O}\big(2^{|Q|^2 \log^2 |Q|}\big)$ if deterministic;*

*that, on any string $w\$$, recognizes whether $w \in L$ when unrolled to $\lceil \log_2 |w| \rceil$ depth.*

Proof in Appendix C. Theorem 1 reveals that running a transformer to $\Theta(\log n)$ depth on inputs of length $n$ unlocks new power compared to a fixed-depth transformer, assuming $\mathsf{TC}^0 \neq \mathsf{NC}^1$. If we do not care that the construction is uniform across layers, we can simplify 8-layer block that determines activeness to 1 layer: we simply hardcode the layer index $\ell$ and use a single transformer layer to compute $i \bmod \ell$. Thus, the non-uniform construction results in a shallower transformer family:

---

[6]Saunshi et al. (2025, Theorem 5.1) also give a log-depth transformer construction for the regular language recognition problem. It is, however, not fully uniform as positions are encoded with vectors of length $\mathrm{O}(\log n)$.

**Corollary 1.1** (Regular Language Recognition, Non-Uniform). *Let $L$ be a regular language over $\Sigma$ and $\$ \notin \Sigma$. There exists a family of causally masked transformers $\{T_n\}_{n=1}^{\infty}$ where $T_n$ has $4\lceil \log_2 n \rceil + 5$ layers such that, on any string $w\$$ of length $n$, $T_n$ recognizes whether $w \in L$.*

These results can be extended beyond regular languages: if a $b$-layer transformer can perform some binary associative operation $\oplus : X \times X \to X$, then one can construct an $\Theta(b \log n)$ layer transformer that computes the iterated version on $n$ values, $x_1 \oplus x_2 \oplus \ldots \oplus x_n \in X$. One example is **iterated matrix multiplication**. For matrices from a fixed set (e.g., $k \times k$ boolean matrices), Theorem 1 already shows that this task can be performed. However, if the matrices are not from a fixed set (e.g., matrices over $\mathbb{Z}$ or $\mathbb{Q}$ or whose shape depends on $n$), then it is unclear whether log-depth transformers can solve the binary multiplication problem, and thus whether they can solve the iterated version.

**Fixed Depth and Bounded Length Inputs.**  Interestingly, while Theorem 1 and Corollary 1.1 are about log-depth transformers, they can be turned around to infer bounds on the input length up to which *fixed depth* transformers (i.e., depth fixed w.r.t. input length) can recognize regular languages. Specifically, given any regular language $L$ and a fixed $d$, Corollary 1.1 implies that there exists a depth $d$ transformer that can recognize strings $w \in L$ as long as $4\lceil \log_2 |w| \rceil + 5 \leq d$,[7] which is satisfied if $4\left(1 + \log_2 |w|\right) + 5 \leq d$, i.e., $|w| \leq 2^{(d-9)/4}$:

**Corollary 1.2** (Depth Scaling for Regular Language). *Let $L$ be a regular language over $\Sigma$ and $\$ \notin \Sigma$. For any $d \in \mathbb{N}$, there exists a causally masked $d$-layer transformer that, on any string $w\$$ of length at most $2^{(d-9)/4} + 1$, recognizes whether $w \in L$.*

An analogous result holds for universal (i.e., shared parameter) transformers from Theorem 1.

## 5   Log Depth Enables Graph Connectivity

In the **graph connectivity problem** (also referred to as STCON or the **reachability problem**), the input is a graph $G$, along with a source vertex $s$ and a target vertex $t$. The task is to determine if $G$ has a path from $s$ to $t$. This is a core problem at the heart of many computational questions in areas as diverse as network security, routing and navigation, chip design, and—perhaps most commonly for language models—multi-step reasoning. This problem is known to be complete for the class of logspace Turing machines (Reingold, 2008; Immerman, 1998), which means that, under common complexity conjectures, it cannot be solved accurately by fixed-depth transformers, which can only solve problems in the smaller class $\mathsf{TC}^0$. However, graph connectivity can be expressed by log-depth *threshold* circuits ($\mathsf{TC}^1$, Barrington and Maciel, 2000), which opens up a natural question: *Can log-depth transformers, which are in $\mathsf{TC}^1$, solve graph connectivity?*

Sanford et al. (2024) provide results showing that a non-uniform log-depth transformer with arbitrarily powerful feedforward nets can solve variants of graph connectivity. We prove that even a *fully uniform* log-depth transformer can solve graph connectivity (proof sketch below, full proof in Appendix D):

**Theorem 2** (Graph Connectivity). *There exists a $(17, 2, 1)$-universal transformer $T$ with both causal and unmasked heads, fixed model dimension $m$, and fixed feedforward width $w$ that, when unrolled $\lceil \log_2 n \rceil$ times, solves connectivity on (directed or undirected) graphs over $n$ vertices: given the $n \times n$ adjacency matrix of a graph $G$, $n^3$ padding tokens, and $s, t \in \{1, \ldots n\}$ in unary, $T$ checks whether $G$ has a path from vertex $s$ to vertex $t$.*

*Proof Sketch.* We will prove this for the more general case of a directed graph $G$ over $n$ vertices. Let $A \in \{0,1\}^{n \times n}$ be $G$'s adjacency matrix. The idea is to use the first $n^2$ tokens of the transformer to construct binary predicates $B_\ell(i, j)$ for $\ell \in \{0, 1, \ldots, \lceil \log n \rceil\}$ capturing whether $G$ has a path of length at most $2^\ell$ from $i$ to $j$. To this end, the transformer will use the $n^3$ padding tokens to also construct intermediate ternary predicates $C_\ell(i, k, j)$ for $\ell \in \{1, \ldots, \lceil \log n \rceil\}$ capturing whether $G$ has paths of length at most $2^{\ell-1}$ from $i$ to $k$ and from $k$ to $j$. These two series of predicates are computed from each other iteratively, as in standard algorithms for graph connectivity:

$$B_0(i, j) \iff A(i, j) \vee i = j \tag{1}$$
$$C_{\ell+1}(i, k, j) \iff B_\ell(i, k) \wedge B_\ell(k, j) \tag{2}$$
$$B_{\ell+1}(i, j) \iff \exists k \text{ s.t. } C_{\ell+1}(i, k, j) \tag{3}$$

---

[7]The inequality holds since Corollary 1.1 generalizes to show $T_n$ recognizes all strings of length $m \leq n$.

The crucial part is to construct a transformer that correctly operationalizes the computation of predicates $B_\ell$ and $C_\ell$. The input to the transformer is the adjacency matrix $A$ represented using $n^2$ tokens from $\{0, 1\}$, followed by $n^3$ padding tokens $\square$, and finally the source and target nodes $s, t \in \{1, \ldots, n\}$ represented in unary notation using special tokens $a$ and $b$:

$$A_{1,1} \ldots A_{1,n} \, A_{2,1} \ldots A_{2,n} \, \ldots \ldots \, A_{n,1} \ldots A_{n,n} \, \underbrace{\square \ldots \ldots \ldots \square}_{n^3} \, \underbrace{a \ldots \ldots a}_{s} \, \underbrace{b \ldots \ldots b}_{t}$$

Let $N = n^2 + n^3 + s + t$, the length of the input to the transformer. The first $n^2$ token positions will be used to compute predicates $B_\ell$, while the next $n^3$ token positions will be used for predicates $C_\ell$.

Initial Layers. The transformer starts off by using layer 1 to store $1/N, n, n^2, s$, and $t$ in the residual stream at every position. It then uses the next 15 layers to compute and store in the residual stream the semantic "coordinates" of each of the first $n^2 + n^3$ token position, namely, $(i, j)$ for each of the first $n^2$ positions $p = in + j$ and $(i, k, j)$ for each of the next $n^3$ positions $p = n^2 + (in^2 + kn + j)$. Finally, layer 17 of the transformer computes the predicate $B_0(i, j)$ at the first $n^2$ token positions.

Repeated Layers. The repeated layers alternate between computing the $C_\ell$ and the $B_\ell$ predicates for $\ell \in \{1, \ldots, \lceil \log n \rceil\}$. The idea is to compute $C_\ell(i, k, j)$ in the $n^3$ padding tokens by attending to positions $(i, k)$ and $(k, j)$ and retrieving $B_{\ell-1}(i, k)$ and $B_{\ell-1}(k, j)$. Similarly, $B_\ell(i, j)$ is computed in the $n^2$ input positions via uniform attention over padding positions $(i, k', j)$ that store $C_\ell(i, k', j)$.

Final Layers. Finally, in layer $2\lceil \log n \rceil + 18$, the final token uses a head that attends with query $\langle \phi(s), \phi(t) \rangle$ corresponding to the source and target nodes $s$ and $t$ mentioned in the input, attending solely to the position with coordinates $(s, t)$, and retrieving the final value $B_{\lceil \log n \rceil}(s, t)$. $\qquad \square$

Thus, while $\mathsf{NC}^1$ circuits (which have log depth) cannot solve graph connectivity unless $\mathsf{NC}^1 = \mathsf{NL}$, log-depth transformers can.

**Fixed Depth and Bounded Length Inputs.** As for regular languages, this result also provides a concrete input length bound up to which a fixed-depth transformer can solve this problem, namely when $18 + 2\lceil \log_2 n \rceil \le d$, which is satisfied if $18 + 2(1 + \log_2 n) \le d$, i.e., $n \le 2^{(d-20)/2}$:

**Corollary 2.1** (Depth Scaling for Graph Connectivity). *For any $d \in \mathbb{N}$, there exists a $d$-layer transformer with both causal and unmasked heads that, on any graph with at most $2^{(d-20)/2}$ vertices, solves the connectivity problem.*

## 6 Comparing Scaling Depth to Scaling Width or Chain of Thought

Our results show that looping layers enables transformers to solve problems likely (conditionally under the conjecture that $\mathsf{TC}^0 \ne \mathsf{NC}^1$) outside $\mathsf{TC}^0$. We now consider how looping compares in expressive power to other ways to add computation to transformers. Rather than increasing depth by repeating layers, one can increase a transformer's *width* via a larger model dimension (Definition 1) or padding tokens (Pfau et al., 2024). Whereas slightly increasing depth likely expands expressive power beyond $\mathsf{TC}^0$, we show that achieving expressivity beyond $\mathsf{TC}^0$ via width likely requires *superpolynomial* width, which is intractable. In contrast to repeating layers, another way to extend inference-time computation is using chain-of-thought (CoT) steps. We thus compare the expressive power achieved repeated layers with CoT steps.

**Wide Transformers with Fixed Depth Remain in $\mathsf{TC}^0$.** Our Corollaries 2.1 and 1.2 show that minimally growing a transformer's depth allows it to express key problems that are likely outside $\mathsf{TC}^0$. In contrast, Theorem 3 (which extends Merrill and Sabharwal (2023a); for completeness, Appendix E gives a sketch) shows that, if depth remains fixed, width must increase *drastically* with sequence length to enable expressive power outside $\mathsf{TC}^0$.

**Theorem 3** (Width Scaling). *Let $T$ be a fixed-depth transformer whose width (model dimension or padding tokens; Pfau et al., 2024) grows at most polynomial in $n$ and whose weights on input length $n$ (to accommodate growing width) are computable in $\mathsf{L}$. Then $T$ can be simulated in $\mathsf{L}$-uniform $\mathsf{TC}^0$.*

Thus, to solve reasoning problems outside $\mathsf{TC}^0$ over a context length $n$, growing depth is much more efficient than growing width. Of course, there may be other types of problems (e.g., those that are

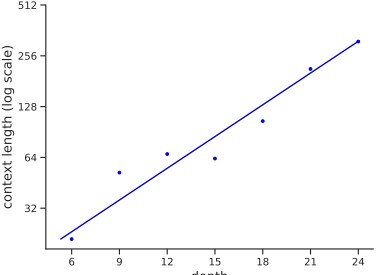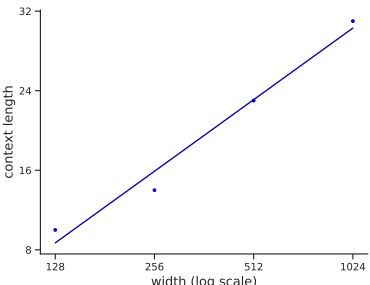

Figure 2: Strong linear fits imply theory/experiment match for modeling the impact of **depth** (left, $d = 4.8 \log_2 n - 15.8$ with $r^2 = 0.93$) and **width** (right, $n = 7.2 \log_2 w - 41.7$ with $r^2 = 0.98$) on effective context length for the $A_5$ state tracking task, a canonical hard regular language recognition problem. As predicted by Theorems 1 and 3, to recognize strings of length $n$, depth only needs to increase minimally $\propto \log n$ while width must increase drastically as $\exp(\Theta(n))$.

knowledge intensive or very parallelizable) where growing width might be more important than growing depth. Petty et al. (2024) provide an interesting empirical investigation of this choice on language modeling, semantic parsing, and other tasks.

**Transformers with Logarithmic Chain-of-Thought Steps Remain in $\mathsf{TC}^0$.** Merrill and Sabharwal (2024, Theorem 4) analyze the power of transformers with $\mathrm{O}(\log n)$ CoT steps, showing it is at most L. However, we have shown that transformers with $\Theta(\log n)$ depth can solve directed graph connectivity, which is NL-complete: this suggests growing depth has some power beyond growing CoT unless L = NL. In fact, the $\mathrm{O}(\log n)$ CoT steps result can be strengthened (Li et al., 2024, Figure 10; for completeness, Appendix E gives a sketch) to an upper bound of $\mathsf{TC}^0$:

**Theorem 4** (CoT Scaling). *Transformers with $\mathrm{O}(\log n)$ chain-of-thought steps can only recognize languages in L-uniform $\mathsf{TC}^0$.*

Thus, while giving a model $\mathrm{O}(\log n)$ CoT steps does not increase its expressive power beyond $\mathsf{TC}^0$, our Theorems 1 and 2 allow $\Theta(\log n)$ to solve key problems that are (likely) outside $\mathsf{TC}^0$. This demonstrates an advantage of dynamic depth over CoT as a form of inference-time compute for reasoning problems including regular language recognition and graph connectivity. It would be interesting to explore this comparison more generally for other problems.

## 7 Experiments: Learning to Recognize Regular Languages

Our theory characterizes the depth and width required to *express* regular language recognition and graph connectivity. Specifically, Theorem 1 predicts that recognizing regular languages over strings of length $n$ is empirically possible with depth proportional to $\log n$. On the other hand, Theorem 3 predicts that the width would need to scale superpolynomially. Here, we aim to empirically measure how much depth and width transformers require in practice when *trained* to recognize regular languages. We will find that expressibility and learnability are highly aligned here: transformers with log depth can learn to recognize regular languages, whereas width must increase superpolynomially with $n$. Moreover, we can empirically quantify the constant factors in these relationships.

We report on an extensive set of experiments to address these questions, training models of different depths and widths on the $A_5$ state tracking task (Merrill et al., 2024), which is a canonical testbed for hard regular language recognition (Theorem 1). The input to the task is a sequence of elements in $A_5$ (the group of even permutations over 5 elements), and the label at each token is the cumulative product of previous permutations up to and including that token (which is itself an element of $A_5$).

We train several (non-universal) transformers with the same architecture used by Merrill et al. (2024) on 100 million $A_5$ sequences of varying lengths up to 1024. To understand the impact of depth and width in a controlled way, we train two series of transformers: the first with width fixed to 512 and depth varying in $\{6, 9, 12, 15, 18, 21, 24\}$, and the second with depth fixed to 6 and width varying

in $\{128, 256, 512, 1024\}$. See Appendix F for further details about our training procedure. After each model is trained, we measure accuracy at each token index from 1 to 1024 and define $n^*$ as the maximum token index at which the model achieved at least 95% validation accuracy. As we trained several seeds with the same depth and width, we aggregate these results across all models with the same depth and width by taking the best-performing (max $n^*$) model. We then plot $n^*$, which represents the effective context length up to which a model can solve the $A_5$ problem, as a function of either depth or width, holding the other variable fixed. We then evaluate if the predicted theoretical relationships between depth, width, and context length hold via an $r^2$ statistic.

The results are shown in Figure 2. When varying depth (left plot), there is a very strong positive correlation ($r^2 = 0.93$) between model depth (x-axis) and $\log n^*$ (y-axis, log scale), the effective (log) context length till which it can solve problems with high accuracy. When varying width (right plot) there is an even stronger positive correlation ($r^2 = 0.98$) between log width (x-axis, log scale) and $n^*$ (y-axis). These results provide strong empirical support for our theoretical predictions that, to recognize regular languages over strings of length $n$, increasing depth logarithmically in $n$ will suffice (Theorem 1), but width must increase exponentially in $n$ (Theorem 3). Figure 2 also gives us a strongly predictive functional form to quantify the impact of scaling depth or width on the effective context length for regular language recognition. The empirical slope for the depth relationship is is 4.8 layers per log tokens. This is less than the slope of 8 derived for universal transformers in Theorem 1, but slightly greater than the theoretical coefficient of 4 for transformers whose depth grows non-uniformly with context length. Thus, our transformers have learned a construction whose depth coefficient is comparable to what we showed was possible in theory, though perhaps slightly more wasteful than it needs to be. Overall, these empirical results show that, in practice, the impact of depth and width on effective context length for regular language recognition aligns with our theoretical predictions, and we are able to empirically fit the quantitative coefficients in the relationships.

# 8 Conclusion

We have shown that recognizing regular languages and graph connectivity, two key problems inexpressible by fixed-depth transformers, become expressible if the depth of the transformer can grow *very slightly* (logarithmically) with the context length by repeating layers. This implies transformers with fixed depth $d$ *can* solve these problems up to bounded context lengths of $2^{O(d)}$. Further, we showed that scaling depth to solve these problems is more efficient than scaling width (which requires superpolynomial increase) or scaling chain-of-thought steps (which requires superlogarithmic increase). As dynamic test-time compute methods have become popular for building more powerful reasoning models such as OpenAI o1 (OpenAI, 2024) and DeepSeek-R1 (DeepSeek AI, 2025), it would be interesting to explore whether universal transformers can realize this theoretical efficiency to provide more efficient long-context reasoning than chain-of-thought steps in practice.

While growing depth enables transformers to solve some key problems outside $\mathsf{TC}^0$, there are limitations on the types of problems log depth should enable solving. Unless $\mathsf{NC} = \mathsf{P}$, log-depth (or even polylog-depth) transformers cannot express $\mathsf{P}$-complete problems including solving linear equalities, in-context context-free language recognition, circuit evaluation, and determining the satisfiability of Horn clauses (Greenlaw et al., 1991). In future work, it would interesting to study the depth required for transformers to solve other interesting problems in $\mathsf{NC}$ including context-free recognition (generalizing regular languages; Theorem 1), which is in $\mathsf{AC}^1$ (Ruzzo, 1981; Venkateswaran, 1991) and boolean formula evaluation, which is $\mathsf{NC}^1$-complete (Buss, 1987). This would help us better understand the degree to which repeating layers can be used as a form of interence-time compute.

## Limitations

We have given constructions whereby looped transformers can express $\mathsf{NC}^1$-hard problems, though we have not considered looped transformers' inductive biases and learning dynamics, which are also important in practice beyond expressivity. Saunshi et al. (2025) empirically study the inductive biases of looped transformers, suggesting looped transformers may generalize in ways favorable for reasoning tasks. Our experiments in Section 7 were conducted with non-looped transformers. It would be interesting in future work to evaluate how the depth requirements change if we enforce that the transformer must re-use the same weights across layers.

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

# A Building Blocks

## A.1 Residual Stream Storage Interface

Our masked pre-norm transformer architecture always normalizes values when reading them from the residual stream. This means that it's not always the case that what's added to the residual stream by one layer is accessible as-is in future layers, which can be problematic if there is a need to "erase" that value. We discuss how values are stored and, if needed, erased from the stream.

For any general scalar $z$, storing $z$ in the residual stream results in $\text{sgn}(z)$ being retrieved when masked pre-norm is applied to this cell. This will be useful when we want to collapse multiple values or perform equality or threshold checks. As a special case, when $z \in \{-1, 0, 1\}$, the retrieved value after masked pre-norm is precisely $z$. Thus scalars in $\{-1, 0, 1\}$ can be stored and retrieved without any information loss.

In order to retrieve a value $z$ with masked pre-norm (rather than just its sign), we can instead represent $z$ as a 4-dimensional vector $\psi(z) = \langle z, 1, -z, -1 \rangle$. Then, pre-norm masked to only this vector will return $\phi(z) = 1/\sqrt{2} \cdot \psi(z)/\sqrt{z^2 + 1}$. Scalars $z$ stored as $\psi(z)$ or $\phi(z)$ in the residual stream can be trivially retrieved as $\phi(z)$ by masked pre-norm:

**Lemma 2.** *There exists a masked pre-norm $\nu$ such that, if $\phi(z)$ or $\psi(z)$ is stored in $\mathbf{h}$, $\nu(\mathbf{h}) = \phi(z)$.*

Furthermore, a single masked pre-norm can even be used to retrieve multiple scalars stored in the residual stream. Since $\phi(z)$ is a unit-norm vector, this is a consequence of the following lemma:

**Lemma 3.** *There exists a masked pre-norm $\nu$ such that, if $\mathbf{h}$ stores unit-norm vectors $\phi_1, \ldots, \phi_k$, then $\nu(\mathbf{h}) = \langle \phi_1, \ldots \phi_k \rangle$.*

*Proof.* We apply the mask to focus on the positions where $\phi_1, \ldots, \phi_k$ are stored. Then, the masked pre-norm outputs

$$\frac{1}{\sqrt{2k}} \langle \phi_1, \ldots, \phi_k \rangle.$$

We can hardcode the scalar multiplier of layer-norm to remove the scalar factor, or, equivalently, bake it into the next linear transformation. Either way, we are able to retrieve the concatenation of $\phi_1, \ldots, \phi_k$ as input to the layer. $\square$

The following establishes that we can compute numerical values $z$ with attention heads and make them accessible as $\phi(z)$ in later layers:

**Lemma 4.** *Let $z$ be a scalar computable by an attention head from residual stream $\mathbf{h}$. There exist two layers producing residual streams $\mathbf{h}', \mathbf{h}''$ such that*

1. *$\phi(z)$ can be read via masked pre-norm from $\mathbf{h}'$ or $\mathbf{h}''$.*
2. *$\phi(z)$ is stored in $\mathbf{h}''$ at (formerly blank) indices $I$.*

*Proof.* The first layer computes $z$ and stores $\psi(z)$ at blank indices $I$ in the residual stream, producing $\mathbf{h}'$. Thus, the second layer can read $\phi(z)$ with masked pre-norm via Lemma 2 and can also recompute $z$ from $\mathbf{h}$, which is a subspace of $\mathbf{h}'$. At this point, it outputs $-\psi(z) + \phi(z)$ at indices $I$, which leads to $\mathbf{h}''$ storing $\phi(z)$ at $I$. $\square$

## A.2 Clearing Stored Values

In the repeated layers of a universal transformer, we will need to overwrite the values stored at particular indices in the residual stream. That is, if $[\mathbf{h}]_I = \mathbf{x}$, it will be useful to produce $\mathbf{h}'$ such that $[\mathbf{h}']_I = \mathbf{y}$ instead. The following lemmas will help implement constructions of this form.

**Lemma 5.** *If a unit-norm vector $\phi$ is stored in $\mathbf{h}$ at $I$, there exists a feedforward sub-layer that removes $\phi$, i.e., produces $\mathbf{h}'$ such that $[\mathbf{h}']_i = \vec{0}$.*

*Proof.* The layer reads $\phi$ via masked pre-norm and writes $-\phi$ to $\mathbf{h}$ at $I$, setting $[\mathbf{h}]_I = \phi - \phi = \vec{0}$. $\square$

Combining Lemma 5 with a parallel layer that stores some new value at $I$, we see that we can effectively *overwrite* values at $I$ rather than just deleting them.

It is also possible to remove information that is not a unit-norm vector, although the construction is less direct.

**Lemma 6.** *Let $\delta$ be the output of a transformer layer on $\mathbf{h}$, targeted to indices $I$ at which $\mathbf{h}$ is blank. Then there exists another transformer layer that resets the residual stream $\mathbf{h}' = \mathbf{h} + \delta$ to $\mathbf{h}$.*

*Proof.* The second layer is a copy of the initial layer that considers the subvector $\mathbf{h}$ of $\mathbf{h}'$ as its input and where all signs are flipped. Thus, it outputs $-\delta$, which guarantees that the final residual stream is $\mathbf{h}'' = \mathbf{h} + \delta - \delta = \mathbf{h}$. $\qquad\square$

## A.3 Computing Position Offsets

It will be useful to show how a transformer can compute the position index of the previous token.

**Lemma 7.** *For a fixed value $k$, assume that at each position $i$, a transformer stores $\mathbb{1}[i = 0]$ and $\mathbb{1}[i < k]$ in the residual stream. Then, there exists a layer that adds $\phi(i - k)$ to the residual stream at token indices $i \geq k$.*

*Proof.* We construct two attention heads. The first head is uniform with value $j$ as $\mathbb{1}[j = 0]$. Thus, the head computes $h_1 = 1/i$. The second head is uniform with value $j$ as $\mathbb{1}[j \geq k]$, and thus computes $h_2 = (i - k)/i$. We then use a feedforward layer to compute:

$$\phi(h_1, h_2) = \phi(\frac{i - k}{i}, \frac{1}{i}) = \phi(i - k).$$

The resulting value is then written to the residual stream. $\qquad\square$

The precondition that we can identify the initial token (cf. Merrill and Sabharwal, 2024) is easy to meet with any natural representation of position, including $1/i$ or $\phi(i)$, as we can simply compare the position representation against some constant.

We assume that the positional encodings used by the model allow detecting the initial token (Merrill and Sabharwal, 2024). One way to enable this would simply be to add a beginning-of-sequence token, although most position embeddings should also enable it directly.

## A.4 Equality Checks

We show how to perform an equality check between two scalars and store the output as a boolean.

**Lemma 8.** *Given two scalars $x, y$ computable by attention heads or stored in the residual stream, we can use a single transformer layer to write $\mathbb{1}[x = y]$ in the residual stream. Furthermore, a second layer can be used to clear all intermediate values.*

*Proof.* After computing $x, y$ in a self-attention layer, we write $x - y$ to a temporary cell in the residual stream. The feedforward sublayer reads $\sigma_1 = \operatorname{sgn}(x - y)$, computes $z = 1 - \operatorname{ReLU}(\sigma_1) - \operatorname{ReLU}(-\sigma_1)$, and writes $z$ to the residual stream.

The next transformer layer then recomputes $y - x$ and adds it to the intermediate memory cell, which sets it back to 0. Thus, the output is correct and intermediate memory is cleared. $\qquad\square$

# B Correctness of Division Construction Attention Heads

The proof of Lemma 1 presents the full construction to implement division in a transformer. For space, we omitted a full proofs of correctness for attention heads in the construction, which we now present. We expect some of these techniques could be reused, though we have not stated them as generally as the gadgets in Appendix A.

**Lemma 9** (First Layer of Lemma 1). *Let $h_i^1$ be an attention head computed with query $q_i = \phi(i/m)$ and keys/values $k_j = v_j = \phi(j)$. Let $e_i = \mathbb{I}(h_i^1 = \phi(i/m))$. Then $e_i = 1$ if and only if $i$ is a multiple of $m$. Furthermore, if $e_i = 1$, then $h_i^1 = \phi(i/m)$.*

*Proof.* Suppose first that $i$ is a multiple of $m$. In this case, there exists a position $j^* \leq i$ such that $i = mj^*$, which means the query $q_i = \phi(i/m) = \phi(j^*)$ exactly matches the key $k_{j^*}$. The head will thus return $v_{j^*} = \phi(j^*) = \phi(i/m)$, representing precisely the quotient $i/m$. Further, the equality check will pass, making $e_i = 1$. The layer thus behaves as intended when $i$ is a multiple of $m$.

On the other hand, when $i$ is *not* a multiple of $m$, no such $j^*$ exists. The head will instead attend to some $j$ for which $i \neq mj$ and therefore $\phi(i/m) \neq \phi(j)$, making the subsequent equality check fail and setting $e_i = 0$, as intended. $\qquad\square$

**Lemma 10** (Second Layer of Lemma 1). *Let $h_i^2$ be an attention head computed with $q_i = \langle 1, 1 \rangle$, key $k_j = \langle e_j, [\phi(j)]_0 \rangle$, and value $v_j = h_j^1$. Then $h_i^2 = \phi(\lfloor i/m \rfloor)$.*

*Proof.* By construction, $q_i \cdot k_j = e_j - [\phi(j)]_0$ where $[\phi(j)]_0 = j/\sqrt{2j^2 + 2}$ is the first coordinate of $\phi(j)$. Note that $[\phi(j)]_0 \in [0, 1)$ for positions $j \leq i$ and that it is monotonically increasing in $j$. It follows that the dot product is maximized at the largest $j \leq i$ such that $e_j = 1$, i.e., at the largest $j \leq i$ that is a multiple of $m$. This $j$ has the property that $\lfloor i/m \rfloor = j/m$. Thus, the head at this layer attends solely to this $j$. Recall that the value $v_j$ at this position is $h_j^1 = \phi(j/m) = \phi(\lfloor i/m \rfloor)$. $\qquad\square$

**Lemma 11** (Fifth Layer of Lemma 1). *Let $h_i^5$ be an attention head computed with query $q_i = \langle \phi(\lfloor i/m \rfloor), 1 \rangle$, key $k_j = \langle \phi(\lfloor j/m \rfloor), b_j^1 \rangle$, and value $v_j = 1 - b_j^2$. Then $h_i^5 = \phi(i \bmod m)$.*

*Proof.* The query-key product achieves its upper bound of 2 exactly when two conditions hold: $\lfloor i/m \rfloor = \lfloor j/m \rfloor$ and $b_j^1 = 1$. Thus, the head attends from $i$ to all $j \leq i$ that store the same quotient as $i$ and also have $b_j^1 = 1$. To make this clearer, let's write $i$ as $i = b'm + c'$ for some $c' < m$. The query-key dot product is then maximized precisely at the $c'$ positions $j$ in $\{b'm + 1, b'm + 2, \ldots, b'm + c'\}$, for all of which $\lfloor j/m \rfloor = \lfloor i/m \rfloor = b'$; note that $b'm$ is *not* included in this list as $b_j^1 = 0$ when $j = b'm$. Of these positions, only $b'm + 1$ has the property that the quotient there is *not* the same as the quotient two position earlier, as captured by the value $v_j = 1 - b_j^2$. Thus, the value $v_j$ is 1 among these positions only at $j = b'm + 1$, and 0 elsewhere.

Assuming $m$ does not divide $i$, $c' > 0$ and the head attends uniformly at $c'$ positions, returning $1/c'$ as the head output. By construction, $c' = i - b'm = i \bmod m$. The layer adds the vector $\psi(1, 1/c')$ defined as $\langle 1, 1/c', -1, -1/c' \rangle$ to the residual stream at position $i$. This, when read in the next layer using masked pre-norm, will yield $\phi(1, 1/c') = \phi(c') = \phi(i \bmod m)$.

On the other hand, if $m$ does divide $i$ (which can be checked with a separate, parallel head), we write $\psi(0)$ to the residual stream, which, when read by the next layer, will yield $\phi(0) = \phi(i \bmod m)$. $\qquad\square$

## C   Regular Language Recognition Proof

**Theorem 1** (Regular Language Recognition). *Let $L$ be a regular language over $\Sigma$ recognized by a (non-)deterministic finite automaton with states $Q$. Let $\$ \notin \Sigma$. Then there exists a causally masked $(0, 8, 9)$-universal transformer with*

- *model dimension $m_{\mathrm{NFA}} = \mathrm{O}(|Q|^2)$, or $m_{\mathrm{DFA}} = \mathrm{O}(|Q| \log |Q|)$ if deterministic;*
- *feedforward width $w_{\mathrm{NFA}} = \mathrm{O}\big(2^{|Q|^4}\big)$, or $w_{\mathrm{DFA}} = \mathrm{O}\big(2^{|Q|^2 \log^2 |Q|}\big)$ if deterministic;*

*that, on any string $w\$, recognizes whether $w \in L$ when unrolled to $\lceil \log_2 |w| \rceil$ depth.*

*Proof.* Regular language recognition can be framed as multiplying a sequence of elements in the automaton's transition monoid (Myhill, 1957; Thérien, 1981). It thus suffices to show how $n$ elements in a finite monoid can be multiplied with $\Theta(\log n)$ depth. We show how a log-depth universal transformer can implement the standard binary tree construction (Barrington and Thérien, 1988; Liu et al., 2023; Merrill et al., 2024) where each level multiplies two items, meaning the overall depth is $\Theta(\log |w|)$. We will represent a tree over the input tokens within the transformer. Each level of the tree will take 8 transformer layers. We define a notion of active tokens: at level 0, all tokens are active, and, at level $\ell$, tokens at $t \cdot 2^\ell - 1$ for any $t$ will remain active, and all other tokens will be marked as inactive. As an invariant, active token $i = t \cdot 2^\ell - 1$ in level $\ell$ will store a unit-norm vector $\delta_i^\ell$ that represents the cumulative product of tokens from $i - 2^\ell + 1$ to $i$.

We now proceed by induction over $\ell$, defining the behavior of non-$ tokens at layers that make up level $\ell$. The current group element $\delta_i^\ell$ stored at active token $i$ is, by inductive assumption, the cumulative product from $i - 2^\ell + 1$ to $i$. Let $\alpha_i^\ell$ denote that token $i$ is active. By Lemma 7 we use a layer to store $i - 1$ at token $i$. The next layer attends with query $\phi(i-1)$, key $\phi(j)$, and value $\delta_j^\ell$ to retrieve $\delta_{i-1}^\ell$, the group element stored at the previous token. Finally, another layer attends with query $\vec{1}$, key $\langle \phi(j)_1, \alpha_i^\ell \rangle$, and value $\delta_{j-1}^\ell$ to retrieve the group element $\delta_{j*}^\ell$ stored at the previous active token, which represents the cumulative product from $i - 2 \cdot 2^\ell + 1$ to $i - 2^\ell$. Next, we will use two layers to update $\delta_i^\ell \leftarrow \delta_i^{\ell+1}$ and $\delta_j^\ell \leftarrow \vec{0}$, which is achieved as follows. First, we assert there exists a single feedforward layer that uses a table lookup to compute $\delta_{j*}^\ell, \delta_i^\ell \mapsto d$ such that $d/\|d\| = \delta_{j*}^\ell \cdot \delta_i^\ell = \delta_i^{\ell+1}$. Next, we invoke Lemma 6 to construct a layer that adds $d$ to an empty cell of the residual stream and then another layer that deletes it. This second layer can now read both $\delta_i^\ell, \delta_{j*}^\ell$ and $\delta_i^{\ell+1}$ (from $d$) as input, and we modify it to add $\delta_i^{\ell+1} - \delta_i^\ell$ to $\delta_i^\ell$, changing its value to $\delta_i^{\ell+1}$. Similarly, we modify it to add $-\delta_{j*}^\ell$ to $\delta_{j*}^\ell$ to set it to 0. A feedforward network then subtracts $\delta_i^\ell$ from the residual stream and adds $\delta_i^\ell \cdot \delta_j^\ell$. This requires at most 4 layers.

To determine activeness in layer $\ell + 1$, each token $i$ attends to its left to compute $c_i/i$, where $c_i$ is the prefix count of active tokens, inclusive of the current token. We then compute $\phi(c_i/i, 1/i) = \phi(c_i)$ and store $c_i$ it temporarily in the residual stream. At this point, we use Lemma 1 to construct 7 layers that compute $c_i \bmod 2$ with no storage overhead. The current token is marked as active in layer $\ell + 1$ iff $c_i = 0 \mod 2$, which is equivalent to checking whether $i = t \cdot 2^\ell - 1$ for some $t$. In addition to updating the new activeness $\alpha_i^{\ell+1}$, we also persist store the previous activeness $\alpha_i^\ell$ in a separate cell of the residual stream and clear $c_i$. This requires at most 8 layers.

Finally, we describe how to aggregate the cumulative product at the $ token, which happens in parallel to the behavior at other tokens. Let $\delta_\$^\ell$ be a monoid element stored at $ that is initialized to the identity and will be updated at each layer. Using the previously stored value $i - 1$, we can use a layer to compute and store $\alpha_{i-1}^\ell$ and $\alpha_{i-1}^{\ell+1}$ at each $i$. A head then attends with query $\vec{1}$, key $\langle \phi(j)_1, 10 \cdot \alpha_{i-1}^\ell \rangle$, and value $\langle (1 - \alpha_{j-1}^{\ell+1}) \cdot \delta_{j-1}^{\ell+1} \rangle$. This retrieves a value from the previous active token $j$ at level $\ell$ that is $\delta_j^\ell$ if $j$ will become inactive at $\ell + 1$ and $\vec{0}$ otherwise. Iff $\delta_j^\ell$ is retrieved, a feedforward network subtracts $\delta_\$^\ell$ from the residual stream and adds $\delta_j^\ell \cdot \delta_\$^\ell$. This guarantees that whenever a tree is deactivated, its cumulative product is incorporated into $\delta_\$^\ell$. Thus, after $\ell = \lceil \log_2 |w| \rceil + 1$ levels, $\delta_\$^\ell$ will be the transition monoid element for $w$. We can use one additional layer to check whether this monoid element maps the initial state to an accepting state using a finite lookup table. Overall, this can be expressed with 8 layers repeated $\lceil \log_2 |w| \rceil$ times and 9 final layers (to implement the additional step beyond $\lceil \log n \rceil$).

Finally, we justify the model size $m$ and feedforward width $w$. To represent a nondeterministic monoid element $Q \to Q$, which is defined by a matrix in $\{0,1\}^{|Q| \times |Q|}$, we must store $s_{\text{NFA}} = |Q|^2$ bits. For a deterministic monoid element, we can reduce this to $s_{\text{DFA}} = |Q| \log |Q|$ by storing the index of the unique 1 in each row of the matrix. The model size is then $m = O(s)$, which gives:

$$m_{\text{NFA}} = O(|Q|^2) \qquad\qquad m_{\text{DFA}} = O(|Q| \log |Q|).$$

The feedforward network stores a lookup table enumerating all possible values of $s^2$ bits, returning $s$ bits in each case. This can be represented as long as $m = O(s)$ and $w = O(2^{s^2})$, which gives:

$$w_{\text{NFA}} = O(2^{|Q|^4}) \qquad\qquad m_{\text{DFA}} = O(2^{|Q|^2 \log^2 |Q|}).$$

This finishes the proof. $\qquad\qquad\qquad\qquad\qquad\qquad\qquad\qquad\qquad\qquad\qquad\qquad\qquad\square$

## D  Graph Connectivity Proof

**Theorem 2** (Graph Connectivity). *There exists a $(17, 2, 1)$-universal transformer $T$ with both causal and unmasked heads, fixed model dimension $m$, and fixed feedforward width $w$ that, when unrolled $\lceil \log_2 n \rceil$ times, solves connectivity on (directed or undirected) graphs over $n$ vertices: given the $n \times n$ adjacency matrix of a graph $G$, $n^3$ padding tokens, and $s, t \in \{1, \ldots n\}$ in unary, $T$ checks whether $G$ has a path from vertex $s$ to vertex $t$.*

*Proof.* We will prove this for directed graphs, as an undirected edge between two vertices can be equivalently represented as two directed edges between those vertices. Let $G$ be a directed graph over $n$ vertices. Let $A \in \{0,1\}^{n \times n}$ be $G$'s adjacency matrix: for $i, j \in \{1, \ldots, n\}$, $A_{i,j}$ is 1 if $G$ has an edge from $i$ to $j$, and 0 otherwise.

The idea is to use the first $n^2$ tokens of the transformer to construct binary predicates $B_\ell(i, j)$ for $\ell \in \{0, 1, \ldots, \lceil \log n \rceil\}$ capturing whether $G$ has a path of length at most $2^\ell$ from $i$ to $j$. To this end, the transformer will use the $n^3$ padding tokens to also construct intermediate ternary predicates $C_\ell(i, k, j)$ for $\ell \in \{1, \ldots, \lceil \log n \rceil\}$ capturing whether $G$ has paths of length at most $2^{\ell-1}$ from $i$ to $k$ and from $k$ to $j$. These two series of predicates are computed from each other iteratively, as in standard algorithms for graph connectivity:

$$B_0(i, j) \iff A(i, j) \lor i = j \tag{4}$$
$$C_{\ell+1}(i, k, j) \iff B_\ell(i, k) \land B_\ell(k, j) \tag{5}$$
$$B_{\ell+1}(i, j) \iff \exists k \text{ s.t. } C_{\ell+1}(i, k, j) \tag{6}$$

We first argue that $B_{\lceil \log n \rceil}(i, j) = 1$ if and only if $G$ has a path from $i$ to $j$. Clearly, there is such a path if and only if there is a "simple path" of length at most $n$ from $i$ to $j$. To this end, we argue by induction over $\ell$ that $B_\ell(i, j) = 1$ if an only if $G$ has a path of length at most $2^\ell$ from $i$ to $j$. For the base case of $\ell = 0$, by construction, $B_0(i, j) = 1$ if and only if either $i = j$ (which we treat as a path of length 0) or $A_{i,j} = 1$ (i.e., there is a direct edge from $i$ to $j$). Thus, $B_\ell(i, j) = 1$ if and only if there is a path of length at most $2^0 = 1$ from $i$ to $j$. Now suppose the claim holds for $B_\ell(i, j)$. By construction, $C_{\ell+1}(i, k, j) = 1$ if and only if $B_\ell(i, k) = B_\ell(k, j) = 1$, which by induction means there are paths of length at most $2^\ell$ from $i$ to $k$ and from $k$ to $j$, which in turn implies that there is a path of length at most $2 \cdot 2^\ell = 2^{\ell+1}$ from $i$ to $j$ (through $k$). Furthermore, note conversely that *if* there is a path of length at most $2^{\ell+1}$ from $i$ to $j$, then there must exist a "mid-point" $k$ in this path such that there are paths of length at most $2^\ell$ from $i$ to $k$ and from $k$ to $j$, i.e., $C_{\ell+1}(i, k, j) = 1$ for *some* $k$. This is precisely what the definition of $B_{\ell+1}(i, j)$ captures: it is 1 if and only if there exists a $k$ such that $C_{\ell+1}(i, k, j) = 1$, which, as argued above, holds if and only if there is a path of length at most $2^{\ell+1}$ from $i$ to $j$. This completes the inductive step.

The crucial part is to construct a transformer that correctly operationalizes the computation of predicates $B_\ell$ and $C_\ell$. The input to the transformer is the adjacency matrix $A$ represented using $n^2$ tokens from $\{0, 1\}$, followed by $n^3$ padding tokens $\square$, and finally the source and target nodes $s, t \in \{1, \ldots, n\}$ represented in unary notation using special tokens $a$ and $b$:

$$A_{1,1} \ldots A_{1,n} \; A_{2,1} \ldots A_{2,n} \; \ldots\ldots \; A_{n,1} \ldots A_{n,n} \; \underbrace{\square \ldots\ldots\ldots \square}_{n^3} \; \underbrace{a \ldots\ldots a}_{s} \; \underbrace{b \ldots\ldots b}_{t}$$

Let $N = n^2 + n^3 + s + t$, the length of the input to the transformer. The first $n^2$ token positions will be used to compute predicates $B_\ell$, while the next $n^3$ token positions will be used for predicates $C_\ell$.

**Initial Layers.** The transformer starts off by using layer 1 to store $1/N, n, n^2, s$, and $t$ in the residual stream at every position, as follows. The layer uses one head with uniform attention and with value 1 only at the first token (recall that the position embedding is assumed to separate 1 from other positions). This head computes $1/N$ and the layer adds $\psi(1/N)$ to the residual stream. Note that the input tokens in the first set of $n^2$ positions, namely 0 and 1, are distinct from tokens in the rest of the input. The layer, at every position, uses a second head with uniform attention, and with value 1 at tokens in $\{0, 1\}$ and value 0 at all other tokens. This head computes $n^2/N$. The layer now adds $\psi(n^2/N, 1/N)$, where $\psi(a, b)$ is defined as the (unnormalized) vector $\langle a, b, -a, -b \rangle$. When these coordinates are later read from the residual stream via masked pre-norm, they will get normalized and one would obtain $\phi(n^2/N, 1/N) = \phi(n^2)$. Thus, future layers will have access to $\phi(n^2)$ through the residual stream. The layer similarly uses three additional heads to compute $n^3/N$, $s/N$, and $t/N$. From the latter two values, it computes $\psi(s/N, 1/N)$ and $\psi(t/N, 1/N)$ and adds them to the residual stream; as discussed above, these can be read in future layers as $\phi(s/N, 1/N) = \phi(s)$ and $\phi(t/N, 1/N) = \phi(t)$. Finally, the layer computes $\psi(n^3/N, n^2/N)$ and adds it to the residual stream. Again, this will be available to future layers as $\phi(n^3/N, n^2/N) = \phi(n)$.

The transformer uses the next 15 layers to compute and store in the residual stream the semantic "coordinates" of each of the first $n^2 + n^3$ token position as follows. For each of the first $n^2$ positions

$p = in + j$ with $1 \leq p \leq n^2$, it uses Lemma 1 (7 layers) with $a_i$ set to $p$ and $m$ set $n$ in order to add $\phi(i)$ and $\phi(j)$ to the residual stream at position $p$. In parallel, for each of the next $n^3$ positions $p = n^2 + (in^2 + kn + j)$ with $n^2 + 1 \leq p \leq n^2 + n^3$, it uses Lemma 1 with $a_i$ set to $p$ and $m$ set $n$ in order to add $\phi((i+1)n + k)$ and $\phi(j)$ to the residual stream. It then uses the lemma again (7 more layers), this time with $a_i$ set to $(i+1)n + k$ and $m$ again set to $n$, to add $\phi(i+1)$ and $\phi(k)$ to the residual stream. Lastly, it uses Lemma 7 applied to $\phi(i+1)$ to add $\phi(i)$ to the residual stream.

Layer 17 of the transformer computes the predicate $B_0(i, j)$ at the first $n^2$ token positions as follows. At position $p = in + j$, it uses Lemma 8 to compute $\mathbb{I}(\phi(A(i, j) = \phi(1))$ and $\mathbb{I}(\phi(i) = \phi(j))$; note that $\phi(A(i, j))$, $\phi(i)$, and $\phi(j)$ are available in the residual stream at position $p$. It then uses a feedforward layer to output 1 if both of these are 1, and output 0 otherwise. This is precisely the intended value of $B_0(i, j)$. The sublayer then adds $B_0(i, j)$ to the residual stream. The layer also adds to the residual stream the value 1, which will be used to initialize the boolean that controls layer alternation in the repeated layers as discussed next.

**Repeating Layers.** The next set of layers alternates between computing the $C_\ell$ and the $B_\ell$ predicates for $\ell \in \{1, \ldots, \lceil \log n \rceil\}$. To implement this, each position $i$ at layer updates in the residual stream the value of a single boolean $r$ computed as follows. $r$ is initially set to 1 at layer 8. Each repeating layer retrieves $r$ from the residual stream and adds $1 - r$ to the same coordinate in the residual stream. The net effect is that the value of $r$ alternates between 1 and 0 at the repeating layers. The transformer uses this to alternate between the computation of the $C_\ell$ and the $B_\ell$ predicates.

For $\ell \in \{1, \ldots, \lceil \log n \rceil\}$, layer $(2\ell - 1) + 8$ of the transformer computes the predicate $C_\ell(i, k, j)$ at the set of $n^3$ (padding) positions $p = n^2 + in^2 + kn + j$, as follows. It uses two heads, one with query $\langle \phi(i), \phi(k) \rangle$ and the other with query $\langle \phi(k), \phi(j) \rangle$. The keys in the first $n^2$ positions $q = i'n + j'$ are set to $\langle \phi(i'), \phi(j') \rangle$, and the values are set to $B_{\ell-1}(i', j')$. The two heads thus attend solely to positions with coordinates $(i, k)$ and $(k, j)$, respectively, and retrieve boolean values $B_{\ell-1}(i, k)$ and $B_{\ell-1}(k, j)$, respectively, stored there in the previous layer. The layer then uses Lemma 8 to compute $\mathbb{I}(B_{\ell-1}(i, k) = 1)$ and $\mathbb{I}(B_{\ell-1}(k, j) = 1)$, and uses a feedforward layer to output 1 if both of these checks pass, and output 0 otherwise. This is precisely the intended value of $C_\ell(i, k, j)$. If $\ell > 1$, the layer replaces the value $C_{\ell-1}(i, k, j)$ stored previously in the residual stream with the new boolean value $C_\ell(i, k, j)$ by adding $C_\ell(i, k, j) - C_{\ell-1}(i, k, j)$ to the same coordinates of the residual stream. If $\ell = 1$, it simply adds $C_\ell(i, k, j)$ to the residual stream.

For $\ell \in \{1, \ldots, \lceil \log n \rceil\}$, layer $2\ell + 8$ computes the predicate $B_\ell(i, j)$ at the first $n^2$ positions $p = in + j$, as follows. It uses a head with query $\langle \phi(i), \phi(j) \rangle$. The keys in the second set of $n^3$ positions $q = n^2 + i'n^2 + k'n + j'$ are set to $\langle \phi(i'), \phi(j') \rangle$ (recall that $\phi(i')$ and $\phi(j')$ are available in the residual stream at $q$) and the corresponding values are set to the boolean $C_\ell(i', k', j')$, stored previously in the residual stream. The head thus attends uniformly to the $n$ padding positions that have coordinates $(i, k', j)$ for various choices of $k'$. It computes the average of their values, which equals $h = \frac{1}{n} \sum_{k'=1}^n C_\ell(i, k', j)$ as well as $1/(2n)$ using an additional head. We observe that $h \geq 1/n$ if there *exists* a $k'$ such that $C_\ell(i, k', j) = 1$, and $h = 0$ otherwise. These conditions correspond precisely to $B_\ell(i, j)$ being 1 and 0, respectively. We compute $h - 1/(2n)$ and store it in the residual stream. Similar to the proof of Lemma 8, the feedforward layer reads $\sigma = \text{sgn}(h - 1/(2n))$, computes $z = (1 + \text{ReLU}(\sigma))/2$, and writes $z$ to the residual stream. The value $z$ is precisely the desired $B_\ell(i, j)$ as $\sigma$ is 1 when $h \geq 1/n$ and 0 when $h = 0$. As in Lemma 8, the intermediate value $h - 1/(2n)$ written to the residual stream can be recomputed and reset in the next layer. As before, the transformer replaces the value $B_{\ell-1}(i, j)$ stored previously in the residual stream with the newly computed value $B_\ell(i, j)$ by adding $\psi(B_\ell(i, j) - B_{\ell-1}(i, j))$ to the stream at the same coordinates.

**Final Layers.** Finally, in layer $2\lceil \log n \rceil + 18$, the final token uses a head that attends with query $\langle \phi(s), \phi(t) \rangle$ corresponding to the source and target nodes $s$ and $t$ mentioned in the input; recall that $\phi(s)$ and $\phi(t)$ are available in the residual stream. The keys in the first $n^2$ positions $p = in + j$ are, as before, set to $\langle \phi(i), \phi(j) \rangle$, and the values are set to $B_{\lceil \log n \rceil}(i, j)$ retrieved from the residual stream. The head thus attends solely to the position with coordinates $(s, t)$, and retrieves and outputs the value $B_{\lceil \log n \rceil}(s, t)$. This value, as argued earlier, is 1 if and only if $G$ has a path from $s$ to $t$. $\quad \square$

# E   Proofs for Width Scaling and Chain of Thought Claims

**Theorem 3** (Width Scaling). *Let $T$ be a fixed-depth transformer whose width (model dimension or padding tokens; [Pfau et al., 2024](#)) grows at most polynomial in $n$ and whose weights on input length $n$ (to accommodate growing width) are computable in* L. *Then $T$ can be simulated in* L-*uniform* $\mathsf{TC}^0$.

*Proof.* By assumption, we can construct an L-uniform $\mathsf{TC}^0$ circuit family in which the transformer weights for sequence length $n$ are hardcoded as constants. Next, we can apply standard arguments ([Merrill et al., 2022](#); [Merrill and Sabharwal, 2023a,b](#)) to show that the self-attention and feedforward sublayers can both be simulated by constant-depth threshold circuits, and the size remains polynomial (though a larger polynomial). Thus, any function computable by a constant-depth, polynomial-width transformer is in L-uniform $\mathsf{TC}^0$. □

**Theorem 4** (CoT Scaling). *Transformers with* $\mathrm{O}(\log n)$ *chain-of-thought steps can only recognize languages in* L-*uniform* $\mathsf{TC}^0$.

*Proof.* The high-level idea is that a polynomial-size circuit can enumerate all possible $O(\log n)$-length chains of thought. Then, in parallel for each chain of thought, we construct a threshold circuit that simulates a transformer ([Merrill and Sabharwal, 2023a](#)) on the input concatenated with the chain of thought, outputting the transformer's next token. We then select the chain of thought in which all simulated outputs match the correct next token and output its final answer. The overall circuit has constant depth, polynomial size, and can be shown to be L-uniform. Thus, any function computable by a transformer with $O(\log n)$ chain of thought is in $\mathsf{TC}^0$. □

# F   Experimental Details

**Curriculum Training.**   In early experiments, we found that learning from long $A_5$ sequences directly was infeasible for our transformer models. We hypothesize this was because, unless earlier tokens are predicted correctly, later tokens contribute significant noise to the gradient. In order to make the learning problem feasible, we follow a curriculum training process, first training on $A_5$ sequences of length 2, then length 4, and continuing up to some fixed maximum power $2^i$. We can then measure the maximum $n^* \le 2^i$ such that the model achieves strong validation accuracy, as mentioned in Section [7](#).

**Depth Experiments.**   All depth experiments used a fixed width of 512. For historical reasons, we have slightly different numbers of runs for different experimental conditions, and some of the runs use different batch sizes (64 and 128). We originally ran a single sweep of depths and widths with 5 runs for depths 6, 12, 18, and 24, each using a batch size of 64 and maximum depth of $2^i = 128$. Seeking to clarify the trend between these original data points, we launched 3 additional runs at depths 9, 15, 18, and 21 using a batch size of 128, which anecdotally sped up training without harming final performance. We also observed that the original depth 24 runs were at the ceiling $n^* = 128$, so we launched 3 additional depth-24 runs with a batch size of 128 and $2^i = 512$ (we also used this larger sequence length for all other runs in the second set). In total, this made:

- 5 runs at depths 6, 12, and 8;
- 3 runs at depths 9, 15, 18, and 21;
- 8 runs at depth 24.

**Width Experiments.**   All width experiments used a fixed depth of 6. We launched 5 runs at widths 128, 258, 512, 1024 with the same hyperparameters, each using a batch size of 64 and $2^i = 128$.

**Compute.**   Each training run was launched on a single GPU. We estimate that, together, these experiments took about 1000 GPU hours.

**License.**   The codebase of [Merrill et al. (2024)](#), which we used for data generation, has MIT license.

