# OpenReview forum: "A Little Depth Goes a Long Way: The Expressive Power of Log-Depth Transformers"
_NeurIPS.cc/2025/Conference — NeurIPS 2025 poster_

### Official Review · Reviewer_i3BG · 2025-07-01

**Clarity:** 3
**Significance:** 4
**Originality:** 4
**Rating:** 5
**Confidence:** 4

**Summary:**

This paper provides a lower bound on the expressivity of universal transformers, that is, transformers whose intermediate layers can be unrolled to arbitrary depth during inference. The authors provide explicit constructions showing that universal transformers of logarithmic depth are capable of recognising regular languages and solving graph connectivity, two fundamental algorithmic building blocks that are beyond the reach of log-precision fixed-depth transformers (at least when the input length is super-exponential in the depth). This result fundamentally advances our understanding of universal transformers, which have recently found success in medium-scale applications (Geiping et al. 2025).

**Questions:**

- When $z$ is stored as $\psi(z)=<z,1,-z,-1>$, is it then not the case that the value retrieved through layernorm is $\psi(z)/\|\psi(z)\|_2=\psi(z)/\sqrt{(2(z^2+1))}$? Is the $\sqrt2$  factor at all relevant? I assume not.

- Do you believe it is feasible to extend your study such that the $\tau \rightarrow 0$ assumption in the attention softmax is not necessary? Can you comment on what kind of a framework would you need in this case?

- The graph connectivity construction requires a cubic amount of padding tokens. Do you think that it might be possible to reduce this amount of tokens? It is not such a crucial question as we are still dealing with sub-exponential padding, I was just wondering.

**Ethical Concerns:**

["NO or VERY MINOR ethics concerns only"]

**Final Justification:**

The theoretical analysis presented in this paper is thorough, creative, and addresses a very important open problem, that of formalising the expressivity of universal transformers. There has been ample evidence of improved out-of-domain generalisation of such models, making a rigorous, formal understanding their properties a valuable contribution.

**Limitations:**

yes

**Quality:**

4

**Strengths And Weaknesses:**

**Strengths**

- Providing explicit constructions for regular language recognition and graph connectivity in the context of **universal** transformers is indeed, as pointed out by the authors, a significant development compared to previously known length-specific constructions used in non-universal transformers. Such constructions are inherently length-generalisable, and I suspect that we might see a surge of experimental analyses of the generalisation of universal transformers following this theoretical finding.
- The level of technical rigour deserves to be commended. While our intuition might tell us that the expressivity of a transformer respects an exponential relationship with the depth, formalising this concept is not a trivial task. It seems that the analysis can even be extended to other settings, i.e., if we know that a transformer can implement some binary associative operator, we can now rigorously show that unrolling the transformer can implement an iterated version of it.


**Weaknesses**

- In my eyes, a glaring weakness of the paper lies in the following fact. The construction using universal transformers was motivated by the limitations of Bingbin Liu's 2023 work, in which each sequence length gets its own transformer parametrization, yet the authors never experimentally test the effectiveness of a universal transformer trained on a short sequence, unrolled to a deeper network on a longer sequence. The paper already does a lot, I do not expect it to provide extensive experimental evidence as it is not the purpose of this work, but I do believe that this experiment could have strengthened the paper, given that a lot of it was motivated through the above mentioned limitations of Liu et al. 2023.
- It is not entirely clear to me how realistic the setting of average-hard attention is. I believe that ideally, the results would also hold in the setup of standard softmax attention. But of course, I acknowledge that every theoretical study needs its assumptions, and these are the assumptions that we are dealing with here. Hard-attention seems more suitable for circuit-based analysis, whereas the softmax case might ultimately be a signal to noise ratio analysis.

---

> ### Author Rebuttal · Authors · 2025-07-30
>
> Thanks for your review! We take your point that experiments with looped transformers would have been interesting to explore, but we have already invested a significant computational budget (about 1000 GPU hours) into the current regular language recognition experiments and are not in a position to rerun them.
>
> > It is not entirely clear to me how realistic the setting of average-hard attention is.
>
> Regarding the motivation and realism of the *average-hard attention* assumption (i.e., $\tau \to 0$), a central motivation of studying the AHAT model is that, without AHAT, it is unclear whether soft-attention transformers could even implement hard-attention retrieval over long contexts. Over any max context length bound, however, soft-attention transformers can arbitrarily approximate AHAT by multiplying all the weights in a head by a specific $\tau$ (which shrinks as context length bound increases).
>
> Some versions of soft-attention transformers (e.g., if the weights can change with $n$) could thus directly simulate AHAT, but if the weights are fixed wrt $n$, it’s unclear if this simulation is possible (and seems intuitively unlikely). Thus, a central motivation in the AHAT model is to abstract away difficulties transformers might have in implementing hard attention. There is definitely opportunity for future work clarifying the relationship of the AHAT model to practice.
>
> > When $z$ is stored as $\psi(z) = \langle z, 1, -z, -1 \rangle$, …
>
> You are right that there is an additional $\sqrt{2}$ factor with layer-norm – we will be sure to double-check that we have not omitted this anywhere. However, the constant factor is not significant, since it can be hardcoded into the weights and thus essentially ignored.
>
> > The graph connectivity construction requires a cubic amount of padding tokens. Do you think that it might be possible to reduce this amount of tokens?
>
> We don’t quite see how to do this, but that doesn’t mean it’s not possible!

---

> > ### Comment · Reviewer_i3BG · 2025-08-04
> >
> > Thank you very much for your rebuttal. I will maintain my positive score.

---

### Official Review · Reviewer_8iy4 · 2025-07-03

**Clarity:** 4
**Significance:** 3
**Originality:** 4
**Rating:** 5
**Confidence:** 4

**Summary:**

This paper studies the expressivity of universal Transformer (aka. weight-tied / looped).

Theoretically, the paper provides expressivity results: Let $n$ denote the input length. The paper shows that (log-precision, poly-width) Transformers with $\Theta(\log n)$ layers can 1) recognize all regular languages (Thm 1) which is NC$^1$-complete, and 2) solve graph connectivity (Thm 2) which is widely conjectured to be outside NC$^1$.
- Both constructions are uniform with respect to $n$.
- As corollaries, these results also show that the context size processable by a fixed-depth Transformer scales exponentially in the Transformer depth.
- These results indicate that on certain problems, depth-recurrence is more efficient than the two alternatives of increasing expressivity:
  - Using a larger width requires a superpolynomial growth in the width (Thm 3).
  - Using chain of thought requires a superlogarithmic number of steps (Thm 4).

Empirically, the paper provides experiments on non-universal Transformers learning $A_5$. The results suggest that depth-scaling is more efficient than width-scaling: the allowed context size grows exponentially in the depth, but only logarithmically in width.

**Questions:**

- One challenge in using depth-recurrence models in practice is that training can be unstable, and unlike CoT, it's hard to collect intermediate supervision for depth-recurrence models. Do you have insights/suggestions on this?
- The experiments are performed with non-weight-tied models. Do you have comments on the difference in inductive biases these non-weight-tied models have compared to the weight-tied models?

**Ethical Concerns:**

["NO or VERY MINOR ethics concerns only"]

**Final Justification:**

I think the paper answers a timely question of comparing different axes of scaling: scaling the depth (depth recurrence) is provably more efficient than scaling width or using longer CoTs. The theoretical results are also corroborated with empirical findings.
I think is a good paper and recommend acceptance.

**Limitations:**

There is no direct societal impact.

A limitation, as with any expressivity result, is that the increased expressivity does not necessarily translated to better learnability and hence practical performance. On the plus side, the paper has partially addressed this by providing empirical results consistent with the theory, using non-weight-tied Transformers.

**Quality:**

4

**Strengths And Weaknesses:**

Strengths:
- The theoretical claims are valid and novel, and answers a timely question of comparing different axes of scaling: scaling the depth (depth recurrence) is provably more efficient than scaling width or using longer CoTs.
- Further, the comparison is empirically verified.
- The constructions provided in Thm 1 & 2 are both uniform, which is more practically relevant than non-uniform constructions.

Weaknesses: I have no major complaints about the paper.

---

> ### Author Rebuttal · Authors · 2025-07-30
>
> Thank you for your review!
>
> > depth-recurrence models in practice … training can be unstable, and … it's hard to collect intermediate supervision
>
> We agree that both of these details (stability and supervision in practice) going beyond expressivity are definitely important. Since we focused on the theoretical expressivity side here, we don’t have much to say about this, besides noting that there is some empirical evidence that looped transformers can be effective on algorithmic tasks. See for example [1]. We will add some discussion highlighting the learning aspects of looping as a potential limitation of this work.
>
> > The experiments are performed with non-weight-tied models. Do you have comments on the difference in inductive biases these non-weight-tied models have compared to the weight-tied models?
>
> We don’t have any novel insights on this. However, [1] makes several interesting claims about this, suggesting weight tying might imbue an inductive bias towards “reasoning”:
>
> > Claim 2: For language modeling, looped models have an inductive bias towards good reasoning despite having worse perplexity and memorization to an iso-flop non-looped model
>
> > Claim 4: Looping-inspired regularization can leverage this inductive bias towards better reasoning
>
> [1] https://arxiv.org/abs/2502.17416

---

> > ### Comment · Reviewer_8iy4 · 2025-08-01
> >
> > Thank you for the response and the reference!

---

### Official Review · Reviewer_GaAn · 2025-07-06

**Clarity:** 3
**Significance:** 4
**Originality:** 3
**Rating:** 5
**Confidence:** 4

**Summary:**

The paper discusses the representation power of a transformer with a depth that grows on the order of the logarithm of the context length, denoted as n, instead of a constant depth. The paper demonstrates that it is possible to construct a transformer with this logarithmic depth and essentially uniform layers that can solve two fundamental problems: recognizing regular languages and solving graph reachability problems of size n.

The key to this construction is a module that can calculate the result of a small fixed integer modulo operation for a sequence of elements. By computing the position of an element in a sequence using a fixed number of layers, the transformer can look up the corresponding feature at that position.

Once this building block is established, the paper shows that not only can we design a transformer with logarithmic depth to recognize regular languages, but we can also design a uniform transformer, which uses repeated layers with shared weights, to achieve the same task with a small constant factor. If we allow for non-uniform construction, this constant can be reduced even further.

The paper also notes that if we consider scaling the width of the transformer, it would need to be super-polynomial to achieve the same complexity class, a result that aligns with previous findings. Additionally, if we only allow a logarithmic number of chain-of-thought steps, the model cannot solve the problem, suggesting that increasing depth is more computationally efficient than increasing width or chain-of-thought steps for this particular problem.

**Questions:**

Please refer to the weakness section for the major questions.

1. Given that it has been shown in some previous works that linear RNN models or some variants of Transformer models can solve some NC-1 problem with constant depth, will logarithmic depth of such modules potentially be in a larger computational class than softmax Transformer?

**Ethical Concerns:**

["NO or VERY MINOR ethics concerns only"]

**Final Justification:**

This paper is of high theoretical importance in understanding the representation power of Transformer and I don't see any significant issues.

**Limitations:**

yes

**Paper Formatting Concerns:**

I didn't notice any.

**Quality:**

4

**Strengths And Weaknesses:**

Overall, the reviewer believes that this is a strong paper and that the weaknesses are relatively minor.

## Strengths:

1. The paper tackles a theoretically important problem by exploring transformers with depth that grows logarithmically with the input size, moving beyond the common and less realistic assumption of constant depth.

2. The construction of the small integer modular module is theoretically interesting and is clearly presented.

3. The theoretical results align closely with empirical findings, even down to small constants, especially in the case of regular language recognition.

## Weaknesses:

1. The abstract might give the impression that scaling depth is universally more efficient than scaling width or chain-of-thought steps, even though the results are specific to the particular context of regular language recognition.

2. The use of the term "uniform" is not fully clarified. In circuit complexity, "uniform" means that there's a systematic way to generate the circuit parameters for any input size. In this paper, "uniform" seems to refer to having fixed parameters across input lengths / layers, and the reviewer is unsure if this matches the computational theory definition. This could create some confusion about whether previous constructions of [1] are truly non-uniform in the traditional sense. It is also unclear to the reviewer that why the authors believe Transformer can't learn a context length dependent weight as the training is performed on a fixed context length.

3.  The reviewer believes that the paper should discuss its relationship with the closely related or possibly concurrent work [2] which shows that a looped transformer can generate latent thoughts and simulate multiple steps of chain-of-thought reasoning using loops. This related work also presents a parameter-uniform construction, and the reviewer would appreciate a discussion on how the two works relate and potentially differ.

[1] Transformers Learn Shortcuts to Automata

[2] Reasoning with Latent Thoughts on the Power of Looped Transformers

---

> ### Author Rebuttal · Authors · 2025-07-30
>
> Thank you for your review!
>
> > The *abstract might give the impression that scaling depth is universally more efficient than scaling width or chain-of-thought steps, even though the results are specific to the particular context of regular language recognition.
>
> We are unsure which part of the abstract you found to be potentially misleading. We tried to scope any depth vs. width vs. CoT comparisons to the two reasoning problems we consider here. In any case, we are happy to change any specific phrases to address this concern.
>
> > The use of the term *uniform* is not fully clarified. In circuit complexity, "uniform" means that there's a systematic way to generate the circuit parameters for any input size. In this paper, "uniform" seems to refer to having fixed parameters across input lengths / layers, and the reviewer is unsure if this matches the computational theory definition. This could create some confusion about whether previous constructions of [1] are truly non-uniform in the traditional sense. It is also unclear to the reviewer that why the authors believe Transformer can't learn a context length dependent weight as the training is performed on a fixed context length.
>
> Thank you for raising this point - indeed, the sense of uniformity that we mean is stronger than standard uniformity, and we will clarify it appropriately. In circuit complexity, uniform indeed means there is a systematic way to generate the circuit for a certain size. Applying this to transformers, we can say that the parameters are $C$-uniform if the function mapping $n$ to the parameters for a network that processes strings of length $n$ is computable in the class $C$. The sense that we mean with **fully uniform** for a transformer is very strong– that $C$ is the class of *constant functions*, i.e., the parameters of the network do not depend on $n$ at all. We will make sure to clarify this appropriately.
>
> >  relationship with the closely related or possibly concurrent work [2] (reasoning with latent thoughts)
>
> Thank you for bringing the concurrent Saunshi et al. paper on looped transformers to our attention -- we will make sure to cite them and add some comparison. Their Theorem 5.1 addresses the regular language recognition problem, but it is not fully parameter-uniform because the parameters (indeed the embedding dimension) can depend on $n$. Related to this, they also use nonstandard position embeddings in the proof, where each position is represented by its binary encoding, which has $O(\log n)$ length. In contrast, our construction is fully parameter uniform and does not use any nonstandard positional embeddings.
>
> > Given that it has been shown in some previous works that linear RNN models or some variants of Transformer models can solve some $NC^1$ problem with constant depth, will logarithmic depth of such modules potentially be in a larger computational class than softmax Transformer?
>
> Good question! The short answer is that our constructions don’t straightforwardly generalize to linear RNNs because they require using attention to retrieve tokens from arbitrarily far back in the context. It’s unclear whether there is some way that linear RNNs could somehow take advantage of logarithmic depth to gain expressivity via a different construction.

---

> > ### Comment · Reviewer_GaAn · 2025-08-01
> >
> > I would like to thank the authors for the rebuttal and the interesting paper. For the abstract I was referring to this sentence "showing that depth scaling is more efficient than scaling width or chain-of-thought steps". I believe it would be better if there is a quantification to show that this is not universal across tasks.
> >
> > Overall, I really appreciate the paper and would keep my positive rating.

---

> > > ### Author Response · Authors · 2025-08-03
> > >
> > > Thanks for your clarification. We will further qualify the abstract sentence you mentioned in revisions.

---

### Official Review · Reviewer_vLZt · 2025-07-07

**Clarity:** 2
**Significance:** 3
**Originality:** 2
**Rating:** 4
**Confidence:** 3

**Summary:**

The given paper analyzes _universal_ transformers, which include a block of layers that can be repeated based on the input size, and how increasing their depth increases their expressive power. The main theoretical results show that logarithmic depth in the input size is sufficient for universal transformers to recognize _regular languages_ and _graph connectivity_. These results are then compared to increasing the width of fixed-depth transformers as well as transformers with logarithmic chain-of-thought steps, both of which remain in the circuit complexity class $TC^0$. This is followed by an experimental evaluation on a dataset that contains one regular language, where the theoretically proposed scaling laws are empirically verified.

**Questions:**

1. More details on the required size of the transformer: For Theorem 1 and following corollaries, are there any assumptions on the required width for transformers, or is this completely independent on, e.g., the size of the alphabet, the number of states in the finite automaton that captures the language, etc.? On a similar note, what about the width of the feedforward layers, is this also problem independent? And in Lemma 1, are there any assumptions on how many attention heads are needed?
2. Definition of universal transformer: If we consider a (2,1,1)-transformer where $r$ is repeated only one time, then, given definition of $L^{\ell}$, $L^1$ and $L^2$ are $s$-layers 1 and 2, but $L^3$ is $r$-layer $(3 - 2) \mod 1 = 0$; shouldn't the indexing of layers start at 1 as well?
3. Comparison to Liu et al. [1]: The main difference to the transformer architecture used in Liu et al. seems to be the universal part, i.e., sharing weights in a dedicated block. However, in Section 4 it is said that Liu et al. strongly simplify their transformer architecture by assuming (i) "specific, nonstandard" positional encodings, and (ii) removing residual connections. In [1], the authors state that they use "fairly standard positional encodings" and use residual connections (both in p. 4, second paragraph). Could you further address these discrepancies?
4. Depths vs. width scaling: Width can be more easily parallelized in comparison to depth; have you considered the practical implications of depth vs. width scaling and if yes, could you please elaborate why scaling depth is preferable despite not being able to parallelize increased depth?
5. Comparison to Sanford et al. [2]: Could you please put Theorem 2 in context with the results for parallelizable tasks in [2], i.e., Theorem 11 and Theorem 18, which also state that log depth is required for transformers that are able to solve st-connectivity.
6. Experiments: Could you please elaborate on your experimental setup, especially lines 358-362 on how $n^*$ is chosen and why only the best-performing models are presented? And, have you considered experiments on st-connectivity (and perhaps any preliminary results)?

[1] Liu, B., Ash, J.T., Goel, S., Krishnamurthy, A. and Zhang, C., 2023. Transformers Learn Shortcuts to Automata. In The Eleventh International Conference on Learning Representations.

[2] Sanford, C., Fatemi, B., Hall, E., Tsitsulin, A., Kazemi, M., Halcrow, J., Perozzi, B. and Mirrokni, V., 2024. Understanding transformer reasoning capabilities via graph algorithms. Advances in Neural Information Processing Systems, 37.

**Ethical Concerns:**

["NO or VERY MINOR ethics concerns only"]

**Final Justification:**

**Reason for not higher score**: I still think the experimental results are a bit underdeveloped.

**Reason for not lower score**: I think the theoretical contributions of the paper are sufficient to justify its acceptance.

**Limitations:**

Please refer to the *Weaknesses* section.

**Paper Formatting Concerns:**

I have no paper formatting concerns.

**Quality:**

2

**Strengths And Weaknesses:**

**Strengths**: The precise characterization of the expressive power of log-depth transformers is an interesting and significant topic, with potential implications for the understanding and application of, e.g., LLMs. The technical content is presented in a clear and concise manner and the paper is well-written overall. The contributions are, for the most part, put well into context with related work. The setting of _universal_ transformers is particularly interesting, as it imposes a fixed parameter budget while maintaining uniformity (up to unrolling the transformer to a certain depth). Moreover, the theoretical results establish a lower bound on the number of layers needed for a certain context length in real-world models, which can be useful for practitioners.

**Weaknesses**:

*Theory*: While the paper presents interesting theoretical contributions, several aspects remain unclear. Most notably, I have open questions about assumptions underlying the logarithmic depth bounds. It is unclear to me whether the width (i.e., embedding dimension, padding tokens, width of feedforward layers) in, e.g., Theorem 1, is truly independent of the input size. For instance, results in [1] and [2] offer more precise bounds on, e.g., embedding dimension or number of attention heads. This also applies to Theorem 2 in the submission, which requires padding tokens polynomial in the input size.

*Experiments*: The empirical evaluation seems underdeveloped. It appears rushed (e.g., differing numbers of runs and different batch sizes due to "historical reasons") and includes only a single task on regular language recognition. Including another task on, e.g., graph connectivity could substantially strengthen the substitution. Moreover, reporting only the best-performing runs is concerning, and further clarification for this choice would be appreciated. In my opinion, the paper could stand alone as a theoretical work, but if experiments are included, they should be more extensive and systematic to support the theoretical findings robustly (see Question 6 for setup concerns).

*Clarity*: The clarity and mathematical rigor could be improved. The informal writing style makes the paper easy to read but at times imprecise; e.g., vague qualifiers like "likely" (lines 310-312) or ambiguous language in proofs (e.g., what exactly is meant by "baking" a scalar multiplier into a linear transformation in the proof of Lemma 3?). The proof sketch for Lemma 1 is difficult to follow without frequent back-and-forth with the Appendix; it would also be helpful to refer to the Appendix if a proof builds upon results which are not yet introduced (e.g., Lemma 7 in proof of Lemma 1). Furthermore, the repeated use of "it" makes the proof difficult to parse at times. It is also unclear whether the transformer architecture in Lemma 1 refers to a (7,0,0), (0,0,7) or (0,7,0) transformer, and whether repeating layers with shared weights is a fundamental part of the proof, or not. Please find some further minor remarks as follows: Variable $\mathbf{m}$ is not introduced yet in line 128; "The repeated layers alternates" -> "The repeated layers alternate" in line 290, "Superpolynomial in width" -> "Superpolynomial width" in line 311; line 347: "depth must increase exponentially" -> I believe "width" is meant instead of "depth"; and in the proof of Lemma 7, it is unclear what the variable $j$ denotes. It would also be helpful to provide more context on how various circuit complexity classes relate to each other, especially for Section 6.

My current score reflects my concerns about the experimental evaluation and open questions about the theoretical contributions. If the authors are able to address these concerns, I would be willing to increase my score.

[1] Liu, B., Ash, J.T., Goel, S., Krishnamurthy, A. and Zhang, C., 2023. Transformers Learn Shortcuts to Automata. In The Eleventh International Conference on Learning Representations.


[2] Sanford, C., Fatemi, B., Hall, E., Tsitsulin, A., Kazemi, M., Halcrow, J., Perozzi, B. and Mirrokni, V., 2024. Understanding transformer reasoning capabilities via graph algorithms. Advances in Neural Information Processing Systems, 37.

---

> ### Author Rebuttal · Authors · 2025-07-30
>
> Thanks for your review!
>
> > width … is truly independent of the input size
>
> Yes! In fact all parameters (their shape and values) in our construction are fully uniform, meaning they have no dependence at all on the sequence length. Thus, the embedding dimension, feedforward dimension, and number of heads do not depend at all on $n$. We will update the theorem statements to indicate this. For Theorem 2, the number of padding tokens does depend on $n$, but, crucially, this is an inference-time dependence and does not result in different parameters for different values of $n$ (unlike prior constructions of Liu et al [1] and Sanford et al [2]; see below).
>
> > Experiments
>
> We took the max over runs (i.e., the best trained model) because we wanted the results to more carefully target *expressivity* (e.g., ignoring learning difficulties, zooming on what is possible for a transformer to represent). We focused our experiments on the regular language task because it has the nice property that every prefix forms a valid subinstance of the problem: thus, to evaluate the maximum length $n^*$ up to which the model succeeds, we can simply find the first index where the prediction was wrong. This property was very handy in our experimental methodology, and is not satisfied with graph connectivity.
>
> > informal language
>
> There are some points in the paper where we indeed use slightly informal language to make the paper more readable (e.g., “likely” to mean assuming $TC^0 \neq NC^1$). We see your point that this sacrifices precision in certain cases and will address this – most crucially in the proofs. In particular, we will clarify Lemma 7 and the other specific points you brought up (thanks for reading carefully).
>
> > Depths vs. width scaling: Width can be more easily parallelized in comparison to depth; have you considered the practical implications of depth vs. width scaling…
>
> Indeed, scaling width is easier than scaling depth. Our claim is in the context of inherently sequential problems such as regular language recognition and graph connectivity, which we focus on. Our theoretical results show that scaling width, even if easier, fundamentally cannot enable solving these problems (Theorem 3) whereas scaling depth logarithmically via looping layers can (Theorems 1 & 2).
>
> We also note that depth scaling in a looped transformer is an inference-time method: the same trained transformer can be looped more, depending on $n$.  Width scaling, on the other hand, is a training-time method: one must *train a different transformer* for each desired width.
>
> > For Theorem 1 and following corollaries, are there any assumptions on the required width for transformers, or is this completely independent of, e.g., the size of the alphabet, the number of states in the finite automaton that captures the language, etc.? On a similar note, what about the width of the feedforward layers, is this also problem independent? And in Lemma 1, are there any assumptions on how many attention heads are needed?
>
> Good question. The embedding dimension, while independent of input length $n$, does need to be large enough to distinguish different tokens in $\Sigma$ and to encode all the possible states. A conservative way to satisfy this would be to construct it as $\max \{ |\Sigma|, |Q} \}$.
>
> The feedforward network is implementing a finite lookup table from $\Sigma \times Q \to Q$. Thus, its width will have to grow with both the number of states of the minimal DFA and the size of the vocabulary. We have not closely analyzed how many attention heads are needed, but it is fixed independent of the language (i.e., there is some fixed number of heads that would work for *any* regular language, no matter the size of the minimal DFA).
>
> We will clarify this.
>
> > If we consider a (2,1,1)-transformer where $r$ is repeated only one time, then, given definition of $L^\ell$, $L^1$ and $L_2$ are $s$-layers 1 and 2, but $L^3$ is $r$-layer $(3 - 2) \mod 1 = 0$; shouldn't the indexing of layers start at 1 as well?
>
> Thanks for reading carefully and catching this. We’ll update the notation so that the three layer lists (s,r,t) are either all 0-indexed or all 1-indexed.
>
> > Comparison to Liu et al. [1]: The main difference to the transformer architecture used in Liu et al. seems to be the universal part, i.e., sharing weights in a dedicated block. However, in Section 4 it is said that Liu et al. strongly simplify their transformer architecture by assuming (i) "specific, nonstandard" positional encodings, and (ii) removing residual connections. In [1], the authors state that they use "fairly standard positional encodings" and use residual connections (both in p. 4, second paragraph). Could you further address these discrepancies?
>
> * Related to what you said, a major difference is that our construction is *fully parameter uniform* (the parameters are the same for any context length), whereas the Liu et al. construction requires the parameters to change with input length $n$.
>
> * As mentioned on page 44, the Liu et al. construction assumes each index $t$ gets two sinusoidal *position embeddings* $\cos(\pi * t / T) and \sin(\pi * t / T)$. While this somewhat resembles the absolute positional embeddings used in the original transformer, our construction is much more general, since it holds without any positional embeddings (as long as a beginning of sequence symbol is used or the first token can be uniquely attended to). We will clarify this difference in revisions and remove the current vague language.
>
> * Another difference between our construction and theirs is the presence of $layer norm$, which is standard in all practical implementations. As they mention on page 25, Liu et al. omit layer norm from their construction, whereas our construction uses (and in fact highly leverages) layer norm.
>
> * Regarding *residual connections*, it’s straightforward to adapt the Liu et al. construction to work with residual connections in a *non-parameter-uniform* setting. However, with *full parameter uniformity* (like we assume), it becomes technically challenging because looped layer blocks need to reuse the same cells in the residual stream. Dealing with this appropriately (*residual stream management*) is a novel contribution of this paper. We will revise the comparison with Liu et al. to better clarify these points.
>
> > Comparison to Sanford et al. [2]: Could you please put Theorem 2 in context with the results for parallelizable tasks in [2], i.e., Theorem 11 and Theorem 18, which also state that log depth is required for transformers that are able to solve st-connectivity.
>
> A key difference between our construction and that of Sanford et al. is, again, *full parameter uniformity*. Whereas our construction uses fixed embedding dimension (and fixed parameters) for all $n$, Theorem 18 in Sanford et al. uses an embedding dimension that grows as a sublinear polynomial in $n$.
>
> Interestingly, their construction uses a similar amount of “padding” tokens ($n^3$ to $n^4$ tokens, depending on the embedding dimension) to ours ($n^3$), unless the embedding dimension is allowed to grow as at least $\sqrt(n)$, in which case their construction can work without padding. We will add some discussion of this comparison, focusing on the central difference of full parameter uniformity for our construction.

---

> > ### Comment · Reviewer_vLZt · 2025-08-04
> >
> > Thanks for the detailed answers!
> >
> > While I am still not fully convinced about the experimental setup, I think the paper offers a really nice theoretical contribution. I trust the authors to carefully proofread their paper for the camera-ready version and implement their proposed changes. I am in favor of accepting the paper and increased my score to reflect this.

---

### Decision · Program_Chairs · 2025-09-17

**Decision:**

Accept (poster)

**Comment:**

The paper studies the expressivity of weight-tied Transformers of depth scaling logarithmically with the context length $n$. They show that log-precision transformers operating over rationals with $\log n$ depth can recognize all regular languages and solve graph connectivity. They additionally show that depth recurrence is more beneficial than width or CoT where both need super polynomial size to compare.

While we knew from prior work, that logarithmic depth Transformers can solve these tasks, the key contribution of the construction is uniformity, that the weights are not dependent on the context length, which was not true for prior work. Additionally, assumptions on the position encodings are much simpler, since the paper uses causal attention and average max-attention to avoid needing length-dependent encodings. This makes the construction a bit more realistic.

I agree with the reviewers that the paper is a nice addition to the line of work on expressivity of Transformers. However there are some technical concerns with Theorem 4 and 5. As stated, the target class is a log-precision transformer, but the rounding semantics for the forward pass aren’t fully specified (there is no discussion other than a footnote). In particular, with masked pre-LayerNorm (involving a square-root) and averaging-hard attention (summing over an arg-max set), it isn’t clear (1) where rounding happens, (2) how ties are broken after rounding in the attention scores, and (3) how the averaging step is implemented (do we round during the multi-operand sum, or sum exactly then round once?). It is important to clarify this, to make sure your claims actually hold.

Since this issue was only realized during the AC-SAC discussion, we have decided to go for a conditional-accept. We ask the authors to fix this concern and if they are unable to, then withdraw the paper. Additionally, we encourage the authors to improve clarity of exposition, point out the limitations of the lack of weight tying in the experiments, and add comparison to existing work on expressivity of looped Transformers. It would also be beneficial for the readers to (1) add a discussion about the dependence of the width and other parameters on the complexity of the regular language (perhaps in terms of the states of the minimal DFA needed to express it), and (2) more clearly explain the position encoding used, since it right now reads as 'there exists some positional encoding that works' but perhaps your result is stronger (correct me if I am wrong) that 'for any positional encoding function where the first position is distinguished', you can get a valid construction.